# A genome-scale yeast library with inducible expression of individual genes

Yuko Arita[1,2,†], Griffin Kim[3,†], Zhijian Li[1,†], Helena Friesen[1], Gina Turco[3], Rebecca Y Wang[3], Dale Climie[1], Matej Usaj[1], Manuel Hotz[3], Emily H Stoops[3], Anastasia Baryshnikova[3], Charles Boone[1,2,4,*] (iD), David Botstein[3,**] (iD), Brenda J Andrews[1,4,***] (iD) & R Scott McIsaac[3,***] (iD)

## Abstract

The ability to switch a gene from off to on and monitor dynamic changes provides a powerful approach for probing gene function and elucidating causal regulatory relationships. Here, we developed and characterized YETI (Yeast Estradiol strains with Titratable Induction), a collection in which > 5,600 yeast genes are engineered for transcriptional inducibility with single-gene precision at their native loci and without plasmids. Each strain contains SGA screening markers and a unique barcode, enabling high-throughput genetics. We characterized YETI using growth phenotyping and BAR-seq screens, and we used a YETI allele to identify the regulon of Rof1, showing that it acts to repress transcription. We observed that strains with inducible essential genes that have low native expression can often grow without inducer. Analysis of data from eukaryotic and prokaryotic systems shows that native expression is a variable that can bias promoter-perturbing screens, including CRISPRi. We engineered a second expression system, $Z_3EB42$, that gives lower expression than $Z_3EV$, a feature enabling conditional activation and repression of lowly expressed essential genes that grow without inducer in the YETI library.

**Keywords** BAR-seq; CRISPRi; gene overexpression; yeast genomics; yeast mutant array

**Subject Categories** Chromatin, Transcription & Genomics; Methods & Resources

**Mol Syst Biol. (2021) 17: e10207**

## Introduction

With its facile genetics and rapid division rate, the budding yeast *Saccharomyces cerevisiae* has been a leading model system for systematic studies of gene function (Botstein & Fink, 2011). To date, the most common genetic approach for exploring the biological roles of genes is to study phenotypes associated with loss-of-function alleles. The first genome-wide gene deletion strain collection was constructed in budding yeast, enabling a broad range of functional profiling studies (Giaever *et al*, 2002). Because ~ 20% of all genes are essential, causing lethality when deleted in a haploid cell (Giaever *et al*, 2002), exploring loss-of-function of essential genes in this context requires the generation of conditional alleles. Conditional knockdown of gene function using degrons, transcriptional repression, and temperature sensitivity has been employed to investigate the role of essential genes, but each strategy has its own drawbacks, one of the most serious of which is general perturbations to cellular physiology associated with changes of environmental conditions (Kanemaki *et al*, 2003; Mnaimneh *et al*, 2004; Li *et al*, 2011; McIsaac *et al*, 2011; Snyder *et al*, 2019).

Another systems-wide approach for studying gene function is gene overexpression, which can produce gain-of-function phenotypes and be used to study both essential and non-essential genes. Several gene overexpression plasmid collections with genes under the control of their endogenous promoters on high copy plasmids have been constructed (Jones *et al*, 2008; Ho *et al*, 2009). Gene expression in these collections is not conditional, limiting phenotypic analysis and precluding the study of genes whose overexpression inhibits growth. To observe more dynamic overexpression phenotypes, the *GAL1* promoter, a strong inducible promoter that can be easily activated by the addition of galactose to glucose-free culture medium, has been used to construct a number of gene overexpression plasmid collections (Sopko *et al*, 2006; Douglas *et al*, 2012). However, *GAL1* promoter-based overexpression systems also

1 Terrence Donnelly Centre for Cellular and Biomolecular Research, University of Toronto, Toronto, ON, Canada
2 RIKEN Centre for Sustainable Resource Science, Wako, Saitama, Japan
3 Calico Life Sciences LLC, South San Francisco, CA, USA
4 Department of Molecular Genetics, University of Toronto, Toronto, ON, Canada
 *Corresponding author. Tel: +416 946 7261; E-mail: charles.boone@utoronto.ca
 **Corresponding author. Tel: +650 769 5510; E-mail: botstein@calicolabs.com
 ***Corresponding author. Tel: +416 978 8562; E-mail: brenda.andrews@utoronto.ca
 ****Corresponding author. Tel: +650 769 5535; E-mail: r.scott.mcisaac@gmail.com
 †These authors contributed equally to this work

have significant drawbacks, including the cell-to-cell variation of expression associated with replicative plasmids and the requirement of a metabolic signal for activation (Karim *et al*, 2013).

To address issues with yeast gene overexpression systems, we previously engineered a genome-integrated, conditional β-estradiol-inducible gene expression system in budding yeast (McIsaac *et al*, 2013a,b, 2014). In this system, an artificial transcription factor consisting of the modular zinc finger DNA-binding domain, human estrogen receptor, and VP16 activation domain is constitutively expressed. We refer to these artificial transcription factors as "ZEVs". A synthetic promoter, which contains binding sites for ZEV variants, is inserted upstream of an open reading frame, displacing the endogenous promoter. The level of activity of the artificial transcription factor is controlled by β-estradiol concentration and enables regulated expression from the corresponding promoter. Since β-estradiol is not a yeast metabolite or signaling molecule, cellular metabolism is not perturbed (McIsaac *et al*, 2013b). ZEVs have been widely used for basic and applied research, including studies of gene regulatory networks (Hackett *et al*, 2020; Kang *et al*, 2020; Ma & Brent, 2020), individual gene function (Elfving *et al*, 2014; Lyon *et al*, 2016; Weir *et al*, 2017; Tran *et al*, 2018; Kim *et al*, 2019; Smith *et al*, 2020; Wang *et al*, 2020; Kira *et al*, 2021), gene regulation (Carey, 2015; Hendrickson *et al*, 2018a; Schikora-Tamarit *et al*, 2018; Lutz *et al*, 2019; Brion *et al*, 2020; preprint: Leydon *et al*, 2021), metabolic engineering (Liu *et al*, 2020), synthetic biology (Schikora-Tamarit *et al*, 2016; Aranda-Díaz *et al*, 2017; Gander *et al*, 2017; Pothoulakis & Ellis, 2018; Bashor *et al*, 2019; Kotopka & Smolke, 2020; Shaw *et al*, 2019; Yang *et al*, 2019), biocontainment (Agmon *et al*, 2017), living materials (Gilbert *et al*, 2021), high-throughput screening (Younger *et al*, 2017; Staller *et al*, 2018), and they have also been adapted to fission yeast (Ohira *et al*, 2017; Gómez-Gil *et al*, 2020; Nuckolls *et al*, 2020) and *Pichia pastoris* (Perez-Pinera *et al*, 2016).

ZEVs allow the rapid induction of a single gene in any environment, which provides a system for tracking how induction of gene expression is directly linked to a cellular response, something that cannot be achieved with deletion mutants (McIsaac *et al*, 2012; Hackett *et al*, 2020; Kang *et al*, 2020). To generate a systems-level reagent set for molecular and cellular analysis using the ZEV system, we constructed the YETI collection, in which nearly every gene in budding yeast is inducible with $Z_3EV$, a ZEV variant that utilizes the 3-finger, Zif268 zinc finger from mouse to bind DNA. We thoroughly characterize these strains and provide details for researchers interested in using this collection. In total, we integrated a uniquely barcoded β-estradiol-regulated promoter in front of 4,668 non-essential genes and 1,022 essential genes in a heterozygous diploid background (as confirmed by PCR) and recovered 4,655 $Z_3EV$-driven non-essential genes in a haploid background using the synthetic genetic array (SGA) selection system (Tong *et al*, 2001). By combining this collection with automated yeast genetics and dynamic growth profiling, we identified 987 genes whose overproduction reduces cell fitness at higher levels of β-estradiol. Additionally, we identified 46 genes whose expression levels affect fitness in a non-monotonic fashion, demonstrating the utility of this collection for genome-scale exploration of fitness landscapes. While more than half of strains with $Z_3EV$-driven essential genes were not able to grow as haploids in the absence of β-estradiol, another subset was viable even without inducer. These findings motivated us to develop

a second expression system—the $Z_3EB42$ system—which involved re-engineering $Z_3EV$ as well as its target promoter to generate a gene regulation system that gives lower expression and is more extensively repressed in the absence of inducer. Together, the YETI strain collection and the $Z_3EB42$ system provide a comprehensive platform for interrogating yeast gene function and dynamics.

# Results

## A genome-scale collection of inducible alleles

To construct a genome-scale collection of strains expressing β-estradiol-inducible alleles, we first constructed a parental strain expressing the $Z_3EV$ transcription factor. Our diploid parental strain, Y14789, was based on the RCY1972 strain, a derivative of S288C. We chose RCY1972 as it is deleted for the *HIS3* locus, making it compatible with SGA methodology, but is otherwise prototrophic, enabling studies of yeast cell growth and other phenotypes in a variety of conditions (Brauer *et al*, 2008). The strain also carries a functional *HAP1* gene, which encodes a transcription factor that localizes to both the mitochondria and nucleus and is required for regulation of genes involved in respiration and the response to oxygen levels (Gaisne *et al*, 1999). Strains derived from S288C typically carry a Ty1 element insertion in the 3' region of the *HAP1* coding sequence, creating a *HAP1* allele that acts as a null for cytochrome *c* expression and leads to mitochondrial genome instability (Gaisne *et al*, 1999). Previous work has shown that removal of the Ty element, which repairs the *HAP1* gene, increases sporulation efficiency dramatically (Harvey *et al*, 2018). To select for strains that carried a functional *HAP1* gene, the gene encoding the $Z_3EV$ transcription factor was integrated next to a functional *HAP1* together with a *natMX* selectable marker in Y14789. Each strain also carries the SGA marker system (*can1Δ::STE2pr-Sphis5* and *lyp1Δ*), enabling automated, array-based genetics. Combining SGA selection with a nourseothricin selection ensures that haploid YETI strains contain $Z_3EV$ as well as a functional *HAP1* allele. Finally, we engineered a DNA template on which the *URA3* gene is linked to $Z_3pr$ for creating genomically integrated promoter fusions. The components of the β-estradiol gene expression system are outlined in Fig 1A.

To test the β-estradiol concentration- and time-dependent expression of a gene regulated by the $Z_3EV$ transcription factor in our strain background, we first inserted the $Z_3pr$ in front of *GFP* and measured GFP fluorescence intensity by flow cytometry. Expression of the $Z_3pr$-*GFP* reporter gene increased in a concentration-dependent and graded manner (Fig 1B, Appendix Fig S1A). Following removal of β-estradiol, the GFP signal decreased by ~ 50% within 3 h and decreased to near-background levels within 24 h, demonstrating that expression of the GFP reporter gene was dependent on β-estradiol concentration and treatment time (Fig 1C). We also tested whether mutant phenotypes could be complemented by cognate genes expressed from the $Z_3$ promoter. We engineered strains in which *LEU2* or *TPS2* were placed downstream of $Z_3pr$ and in the absence of β-estradiol, the resultant strains displayed known phenotypes associated with *leu2Δ* (Appendix Fig S1B) and *tps2Δ* (Appendix Fig S1C), leucine auxotrophy (Toh-e *et al*, 1980) and heat sensitivity (Gibney *et al*, 2015), respectively, but grew equivalently to WT cells in the presence of β-estradiol.

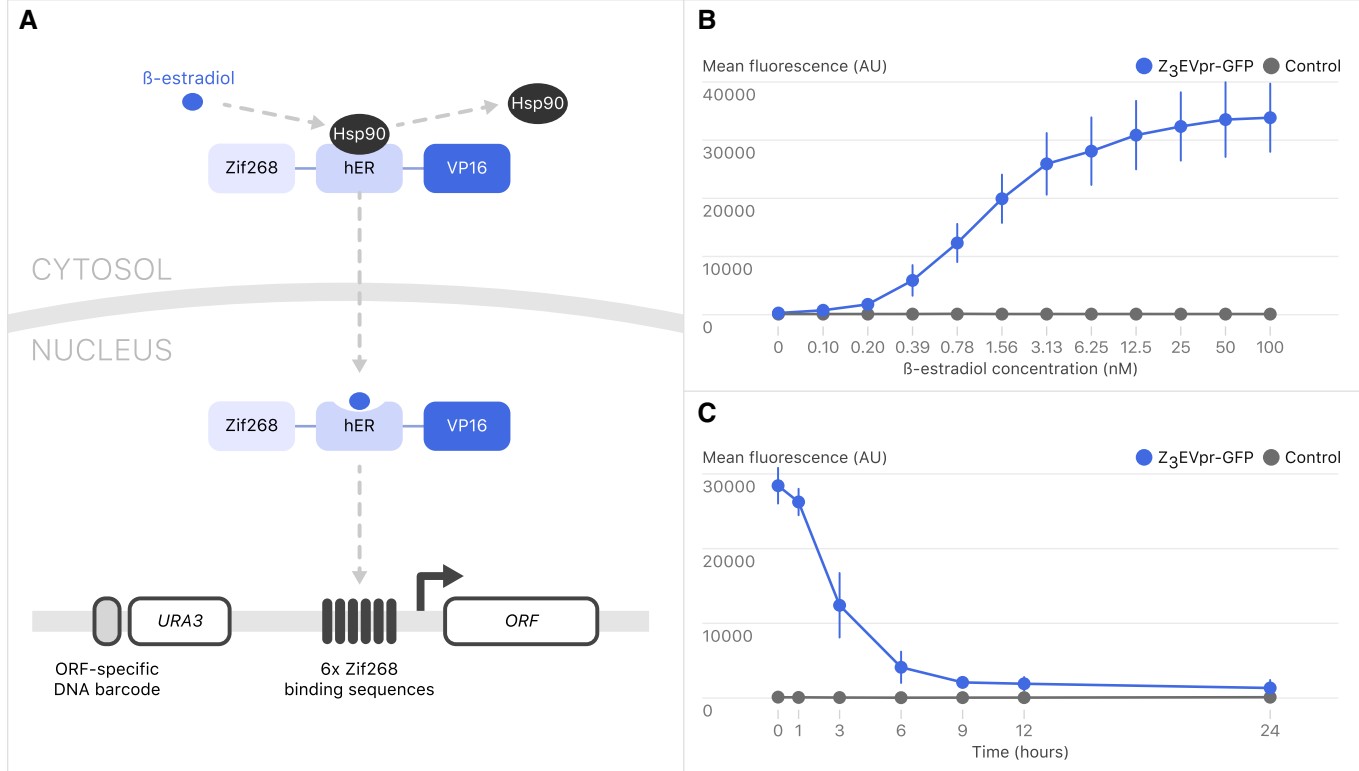

**Figure 1. The Z$_3$EV system.**

A Outline of the β-estradiol-inducible gene expression system. Z$_3$EV is composed of a 3-zinc finger DNA-binding domain (Zif268), human estrogen receptor domain (hER), and transcription activation domain (VP16). β-estradiol displaces Hsp90 from the estrogen receptor, allowing Z$_3$EV to translocate to the nucleus and induce gene expression. Zif268 binds preferentially to a sequence that is present in six copies in Z$_3$pr. In the strain collection, gene-specific DNA barcodes are flanked by universal primer sequences: 5'-GCACCAGGAACCATATA-3' and 5'-GATCCGCTCGCACCG-3'.

B GFP intensity as a function of β-estradiol concentration. Strains with an integrated Z$_3$pr driving GFP (Y15292; blue) and a control strain (Y15483; gray) were incubated with a concentration series of β-estradiol in YNB for 6 h, and then cells were fixed. GFP intensity was measured by flow cytometry. Error bars represent standard deviation for three replicates.

C Y15292 (blue) and Y15477 (gray) cultures were induced with 10 nM β-estradiol for 6 h. Cells were washed, and β-estradiol was removed from the medium at time = 0 h on the figure. Error bars represent ± 1 standard deviation for three biological replicates.

Following these successful characterizations of our constructs, we engineered a genome-wide collection in which the endogenous promoters of individual genes were replaced by inserting the Z$_3$pr just upstream of the start codon of each ORF in a heterozygous diploid strain carrying the Z$_3$EV transcription factor under the control of the constitutive *ACT1* promoter, which we linked to a *natMX* marker. The *URA3* gene marking the Z$_3$ promoter is expressed divergently from the Z$_3$pr-controlled target gene. Importantly, the *URA3* marker gene in each strain is linked to a unique DNA molecular barcode such that the resulting genome-wide β-estradiol-inducible strain collection is compatible with pooled screening approaches (Fig 1A). Promoter insertions were placed directly upstream of the first *ATG* of each *ORF* and did not remove any native DNA. Rather than removing the native promoter sequence from the genome, which we believe is likely to disrupt the expression of neighboring genes, native promoters were simply displaced by ~ 2 kb. Additionally, yeast does not have "transcriptional activation at a distance". From the work of Dobi and Winston (2007), once an activation sequence was placed 690+ bp from a target gene, it was no longer regulatory. Thus, we expect that displacement of the native promoter by ~ 2 kb should be sufficient for removing its regulatory potential. Promoter insertions were confirmed by PCR, and further quality control with a subset of specific strains was carried out using whole-genome sequencing (see Appendix, Dataset EV1). In total, we constructed 1,022 diploid strains carrying Z$_3$pr alleles of essential genes (corresponding to 97.1% of essential genes) (Dataset EV2A) and 4,668 diploid strains expressing Z$_3$pr alleles of non-essential genes (corresponding to 98% of non-essential genes) (Dataset EV2B).

For the non-essential genes, we recovered haploid derivatives by sporulating the heterozygous diploid strains and germinating haploid meiotic progeny on SGA selection medium (Dataset EV2C). We refer to these haploid strains as the YETI non-essential (YETI-NE) panel. The set of strains with essential genes under the control of the β-estradiol-inducible promoter are maintained as diploids. We refer to these strains as the YETI essential (YETI-E) panel. In total, we were unable to construct 127 strains (30 YETI-E diploids, 84 YETI-NE diploids, and 13 YETI-NE haploids; Dataset EV2D).

## Quantifying the relationship between inducer level and transcript level

A useful feature of the $Z_3EV$ system is the potential to either knock down or induce a gene of interest depending on the level of β-estradiol inducer, which enables comprehensive analysis of the relationship between gene expression and phenotype by finely tuning transcription. To assess this property of the collection, we selected $Z_3pr$ alleles of 18 non-essential genes with a range of native expression levels (from 3 to 2,173 TPM [transcripts per million]). Following growth in the presence of various concentrations of β-estradiol for 30 min, we quantified mRNA expression levels of the β-estradiol-regulated genes using RNA-seq. All genes had qualitatively similar responses to β-estradiol concentration: very low expression at β-estradiol concentrations from 0–1 nM and then an increase in expression between 4 and 16 nM, followed by a plateau at β-estradiol concentrations beginning at 16–64 nM (Fig EV1A). However, the actual level of transcript produced at each β-estradiol concentration varied. For this gene panel, maximum expression levels ranged from ~ 30–40 TPM for some genes (e.g., *ATG4*, *SNT1*, and *SRS2*) to over 1,000 TPM for others (e.g., *ASC1*, *BAT1*, *THR1*, and *VMA3*) at > 16 nM β-estradiol. Peak expression levels were correlated with native transcript levels (Pearson correlation = 0.74), which is reminiscent of the gene-dependent expression variation reported with a GALpr plasmid collection (Gelperin *et al*, 2005). For genes in the panel whose native expression was < 250 TPM, the maximum expression output from $Z_3EV$ and the level of native expression followed a simple linear relationship (Fig EV1B). Linearity broke down for the two highly expressed genes we tested, *ASC1* and *VMA3*; even at saturating concentrations of β-estradiol, $Z_3EV$-induced expression was lower than or equal to native expression levels. We conclude that $Z_3EV$ can be used for titrating gene expression, but the exact number of transcripts produced depends on the target gene open reading frame and its genomic context. Additionally, since we used RNA-seq, we explored the transcriptome landscape of these strains. Our most striking observation was that Bat1 induction resulted in the repression of a variety of amino acid biosynthesis genes in a dose-dependent fashion, including all of the *ILV* (IsoLeucine-plus-Valine requiring) genes, which are upstream of Bat1 and are members of the superpathway of branched-chain amino acid biosynthesis (Appendix Fig S2). Since Bat1 catalyzes the terminal reactions in this superpathway, the transcriptional responses are consistent with end-product inhibition.

## Growth patterns associated with β-estradiol-dependent regulation of essential genes

To begin our characterization of the YETI strain collection, we wanted to describe the possible behaviors of the β-estradiol-regulated promoter alleles. We first explored the growth characteristics of the essential gene alleles (YETI-E) using high-resolution time-lapse imaging (see Materials and Methods for details) at twelve β-estradiol concentrations since, unlike non-essential genes, alterations in essential gene expression are expected to lead to an easily assayed growth phenotype (Fig 2). YETI-E strains were grown as heterozygous diploids, sporulated and haploid YETI-E strains were selected on medium with various β-estradiol concentrations. Colony growth was measured over time and growth curves were quantified by determining the area under the growth curve (AUGC) (Fig EV2). We utilized this metric because it is insensitive to specific data parametrizations. Normalized AUGC values were then hierarchically clustered using a Chebyshev distance metric, which revealed five distinct clusters or promoter behaviors (Fig 2, Dataset EV3A). The largest cluster contained 49% of YETI-E strains, each of which showed a dosage-dependent growth response, with no growth in the absence of inducer, and improved growth with increasing β-estradiol concentrations (Cluster 5, Fig 2). A second smaller cluster showed a similar initial behavior, with growth depending on the presence and concentration of inducer, but with growth inhibition at higher β-estradiol concentrations, indicating dosage toxicity (Cluster 4 with 4.7% of strains). Twenty-six out of 46 genes in Cluster 4 have been shown to be toxic upon overexpression in one or more plasmid collections under control of the *GAL1* promoter (Gelperin *et al*, 2005; Sopko *et al*, 2006; Douglas *et al*, 2012). In total, more than half (53.7%) of the YETI-E strains exhibited β-estradiol-dependent growth that is "tunable" by inducer concentration.

Most remaining strains in the YETI-E collection had a dosage-independent growth response, including a large set of strains (33%) that grew well in the absence of β-estradiol (Cluster 1); many of these had a mild growth impairment in high β-estradiol concentrations. A smaller set (9.2%) grew in the absence of inducer, with more substantial growth inhibition at higher concentrations (Cluster 2—dosage toxicity). Finally, a small group of strains (4.2%), with no obvious functional features in common, failed to grow well regardless of the inducer concentration, suggesting these strains have a non-functional promoter (Cluster 3—dosage-independent lethality) (Dataset EV3B).

Roughly half of YETI-E strains grew in the absence of β-estradiol, indicating sufficient basal gene expression to support essential gene function (Fig 2, Dataset EV3A). Our transcriptome analysis of a panel of 18 $Z_3pr$-driven alleles showed that transcript levels were reduced to 30%, on average, of their native levels in the absence of inducer (Appendix Fig EV1A, Appendix Fig S3), and similar results were found with 201 β-estradiol-controlled transcription factors (Hackett *et al*, 2020). The ability of low levels of YETI-E allele expression in the absence of inducer to support growth may correlate with the levels of native gene expression. To test this possibility, we divided essential genes into six bins based on native gene expression levels (Lipson *et al*, 2009) and plotted the growth distribution of each bin. Genes with low transcript levels had

**Figure 2. Hierarchical clustering growth patterns of YETI-E haploid strains.**

The YETI-E diploid strains were sporulated, and haploids were selected on SC SGA selection medium with monosodium glutamate as a nitrogen source at 12 β-estradiol concentrations (0, 0.01, 0.03, 0.1, 0.3, 1, 3, 10, 30, 100, 300, 1,000 nM) and clustered by growth profile. Growth was measured as the area under the growth curve (AUGC); blue represents little growth and yellow represents more growth. AUGC values were quantified for sixteen separate colonies per genotype and dose (see Materials and Methods). Strains were clustered using a Chebyshev distance metric, resulting in five clusters (numbered 1–5). Representative strains for each of the five clusters and their growth patterns are plotted to the right of the clustergram. Error bars represent ± 1 standard deviation.

▶

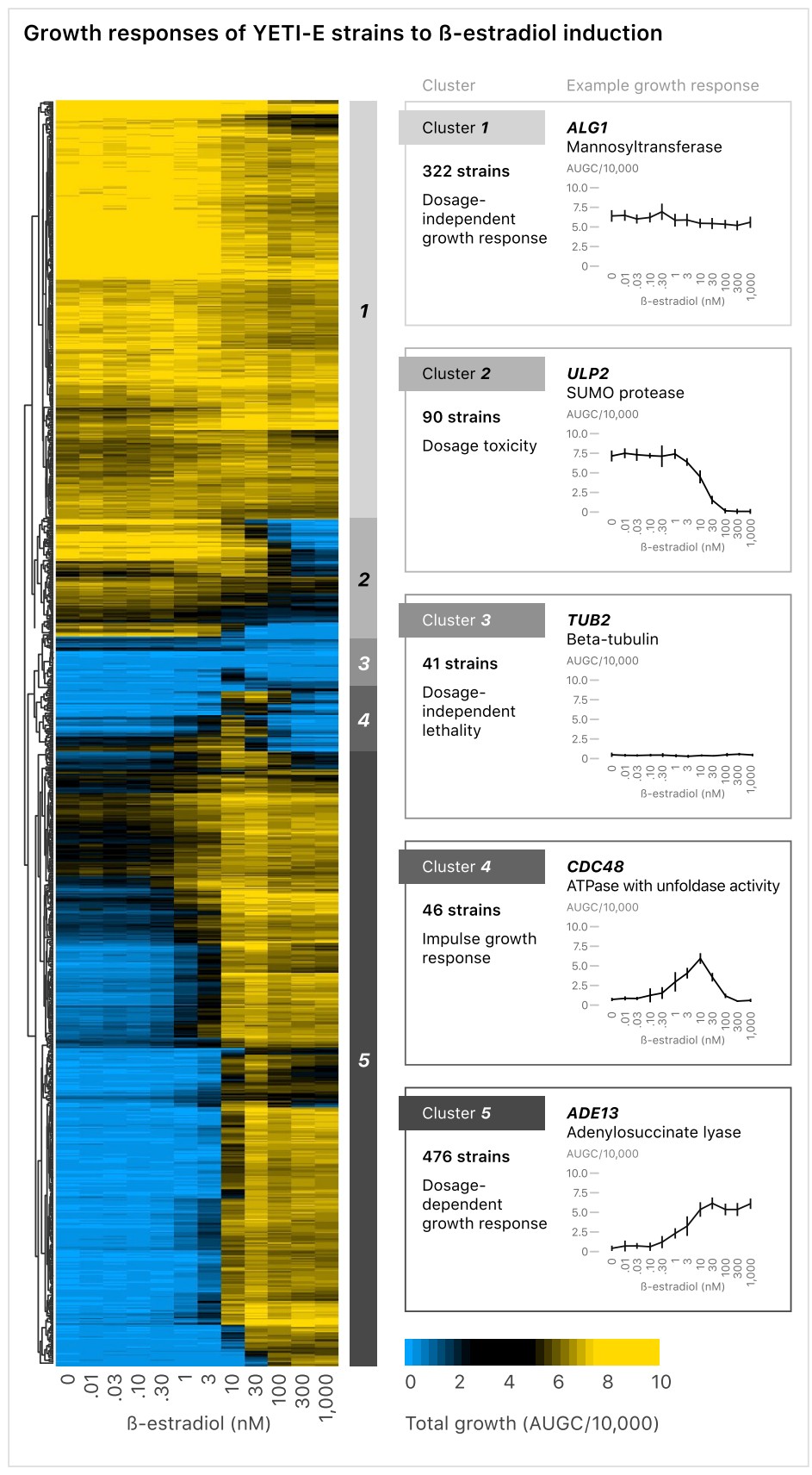

**Figure 2.**

significantly better growth in the absence of β-estradiol than genes with high transcript levels (KS test, *P*-value < $1.9 \times 10^{-20}$; Fig 3A, Dataset EV4A). With unbinned data, the Spearman correlation between transcript level and AUGC (at 0 nM β-estradiol) was −0.49 (*P*-value < $2.2 \times 10^{-16}$). The converse is also true: YETI-E strains that grew poorly at 0 nM β-estradiol contained regulated alleles of genes that were more highly expressed from their native promoters than strains that grew well (Appendix Fig S4). We conclude that though $Z_3$pr is extremely weak in the absence of β-estradiol, it still promotes enough transcription of many essential genes that are normally lowly expressed to facilitate growth.

We compared our results to the growth characteristics and gene expression patterns of strains in other yeast collections that enable conditional induction or repression of essential genes. In the Tet-off collection (Mnaimneh *et al*, 2004), the native promoter of a target gene is replaced by a constitutive promoter that contains elements that allow it to be repressed in the presence of doxycycline. Of the 453 strains tested in both systems, 224 had strongly reduced growth in the presence of doxycycline (dox-responsive) and 229 were either not affected or were minimally affected by doxycycline (dox-non-responsive). We found that 74% of dox-non-responsive genes were also β-estradiol-non-responsive (union of Clusters 1 + 2, Fig 2). In contrast, only 30% of dox-responsive genes were β-estradiol-non-responsive, a ∼ 1.6-fold reduction from what is expected by random chance (Dataset EV4B). Dox-responsive genes were also more highly expressed than dox-non-responsive genes (Fig 3B *P*-value = 0.002, two-sided *t*-test). Thus, the ability to achieve conditional growth with systems that either "turn down" (Tet-off) and "turn up" ($Z_3$EV) gene expression is influenced by native gene expression level.

We used the literature to explore if yeast strains bearing essential genes whose expression was perturbed in other ways had the same inverse correlation between native transcript level and fitness. The DAmP (Decreased Abundance by mRNA Perturbation) collection contains 842 essential genes whose 3′ untranslated region (UTR) is disrupted with an antibiotic resistance cassette, which can destabilize the corresponding transcript (Breslow *et al*, 2008). Fitness of the DAmP strains was assessed using a competitive growth assay. Consistent with our findings with the $Z_3$EV and Tet-off systems, DAmP alleles that gave closer-to-wild-type fitness tended to be in genes with lower native expression levels than DAmP alleles that gave reduced fitness (Fig 3C; Dataset EV4C, *P*-value = 0.002, two-sided *t*-test between the first and tenth decile). N-terminal heat-inducible-degrons can also be used to target a fusion protein for degradation at 37°C in a Ubr1-

dependent manner (Kanemaki *et al*, 2003). Of 94 degron-tagged essential genes, those for which the degron had no effect on temperature sensitivity had lower native expression than those for which the degron caused a temperature-sensitive phenotype (Fig 3D; Dataset EV4D, *P*-value = 0.0001, two-sided *t*-test). Finally, we looked at the effects of a new inducible CRISPR-interference (CRISPRi) library on growth of strains with essential genes (Momen-Roknabadi *et al*, 2020). CRISPRi depletion scores for highly expressed essential genes were significantly stronger (more negative) than those of lowly expressed essential genes (Fig 3E; Dataset EV4E, *P*-value = $3.6 \times 10^{-5}$, two-sided *t*-test between the first and tenth decile).

To test if native expression level is also an important factor for achieving conditional growth for essential genes in organisms other than yeast, we investigated a recent CRISPRi pooled screen from *Escherichia coli* (Wang *et al*, 2018). A useful feature of *E. coli* for this analysis is the availability of a deletion mutant collection from which a core set of essential genes has been determined (Baba *et al*, 2006; Wang *et al*, 2018). In the CRISPRi screen, 62% of strains with essential genes scored as "non-dividing" (Class ND), i.e., these genes were inhibited enough by CRISPRi to block cell division. Thirty-eight percent of essential genes were not inhibited enough by CRISPRi to prevent cells from dividing (Class D). We find that Class ND genes were, on average, 5× more highly expressed than Class D genes (Fig 3F; Dataset EV4F, *P*-value = 6.576e-11, two-sided *t*-test).

Collectively, we find that perturbations of expression of essential genes in eukaryotes and prokaryotes, including promoter replacement ($Z_3$EV and Tet-off), 3'-UTR disruption, N-terminal degron, and CRISPRi, do not have equal effects on fitness for all genes. Reducing expression of essential genes with high native transcript levels tends to cause fitness defects, and genes with low expression are more resistant to the effects of synthetic perturbation.

## Growth profiles associated with β-estradiol-dependent regulation of non-essential genes (YETI-NE)

By definition, failure to express a non-essential gene in a strain grown in a rich medium is not expected to produce a dramatic growth phenotype. However, some non-essential genes are required for growth in certain conditions, including metabolic auxotrophs. We defined metabolic auxotrophs as a set of 80 genes annotated in the *Saccharomyces Genome Database* (SGD) (Cherry *et al*, 2012) that typically function in the biosynthesis of amino acids or nucleotides and whose deletion causes amino acid or nucleotide auxotrophy. Of

**Figure 3.  Synthetic control of growth depends on native gene expression levels in yeast and in *E. coli*.**

A      Yeast strains in which $Z_3$EV targets essential genes are more likely to grow in the absence of inducer if the $Z_3$EV-controlled gene is lowly expressed. The boxplots show YETI-E AUGC (area under the growth curve) values from 0 nM (gray) and 1,000 nM (blue) β-estradiol experiments. The six bins (bins 1–6) are based on gene expression level of native genes (Lipson *et al*, 2009).

B–F    Box plots showing native levels of gene expression binned by growth characteristics for other yeast strain collections. (B) The TET-allele collection. Genes not repressible with Tet-off (DOX-non-responsive) and genes that are repressible with Tet-off (DOX-responsive) are shown. (C) The DAmP allele collection. Box plots showing fitness of DAmP alleles as a function of native expression level. (D) TS-degron strains. Levels of native gene expression for genes for which the addition of an N-terminal degron does not affect growth versus those for which the degron confers temperature-sensitive growth are plotted. (E) CRISPRi strains. Box plots showing fitness of CRISPRi alleles as a function of native expression level. (F) *E. coli* strains in which CRISPRi targets essential genes are more likely to grow if the targeted essential gene is lowly expressed. Boxplots show the distribution of gene expression levels for *E. coli* genes tested in a CRISPRi screening experiment (Wang *et al*, 2018). The genes are grouped into essential genes whose repression by CRISPRi inhibits growth (does not grow) or fails to inhibit growth (grows). Gene expression data from Gene Expression Omnibus (GSE67218). RPKM (Reads Per Kilobase of transcript, per Million reads) median values are 166 and 503 for the "Grows" and "Does not grow" classes, respectively. For all box plots, the central band represents the median value. The bottom and top hinges represent the 25 and 75% quantiles, respectively.

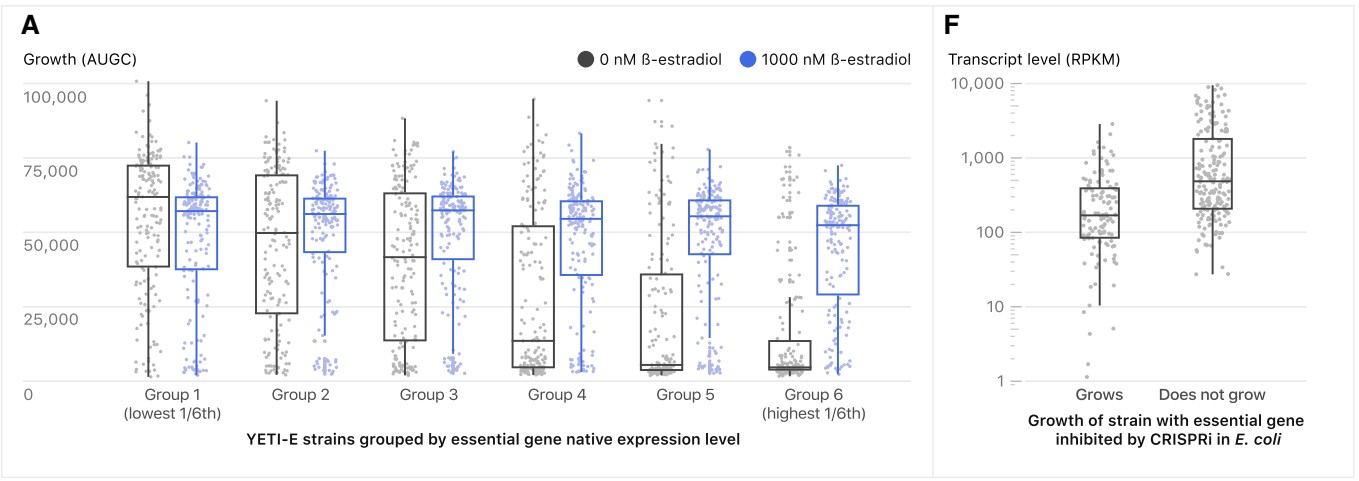

**Native expression is an important variable for growth-based screens in yeast**

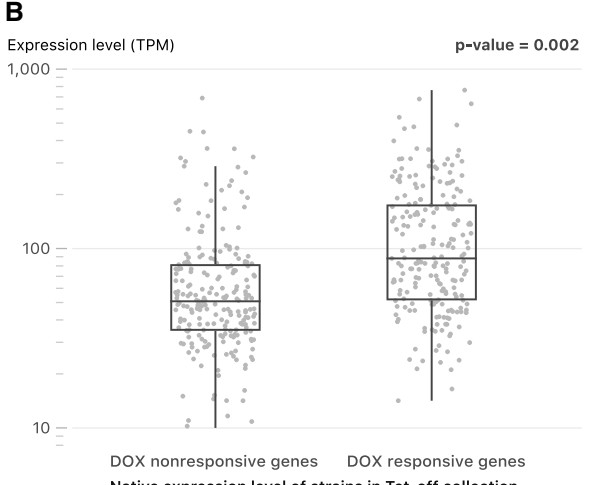

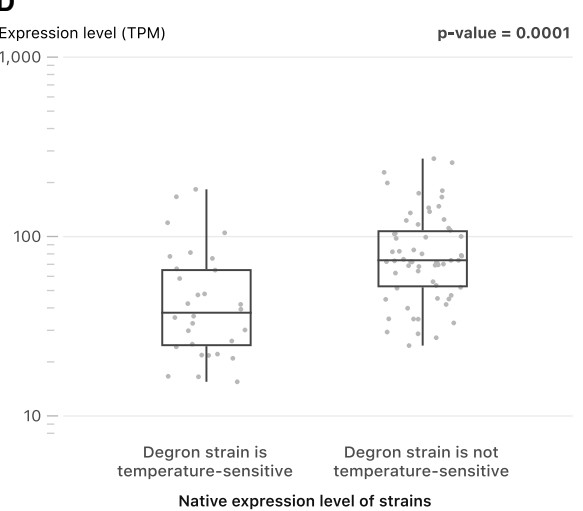

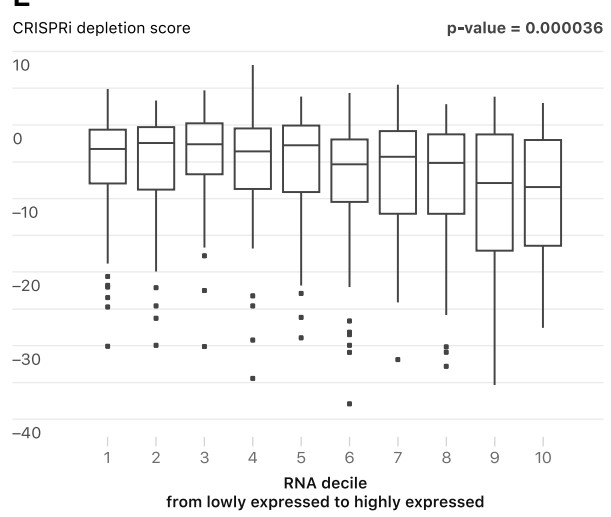

**Figure 3.**

these, 14 metabolic auxotrophs are not present in the YETI-NE collection and 24 metabolic auxotrophs are in pathways that are precluded from our analysis because they are part of the SGA selection protocol used in the construction of the YETI-NE collection (uracil, lysine, arginine, and histidine). Indeed, whole-genome sequencing of a subset of these SGA selection-pathway strains, grown on minimal medium under a condition where they should not actually propagate, revealed chromosome duplication or partial duplication events in 10 of the 17 strains (~ 59%) (Dataset EV1), with most events (8/10) including the gene of interest. For haploid YETI-NE strains *outside* of selection pathways that we sequenced, 4% (1 strain out of 23) had detectable aneuploidy. In total, 42 metabolic auxotrophs in the YETI-NE collection could be assessed for β-estradiol-dependent growth in auxotrophic conditions.

To explore the behavior of the auxotrophic allele panel in the YETI-NE collection, and to reveal any additional auxotrophies, we quantified the growth of the entire YETI-NE haploid panel on minimal YNB (yeast nitrogen base without amino acids or bases) and richer SC (yeast nitrogen base with all amino acids and bases added; see Appendix) solid medium in the absence or presence of β-estradiol (Dataset EV5A). To quantify strain growth, we defined a metric called the Aux score, which provides a measure of whether a non-essential gene behaves like an auxotroph on YNB medium: Aux score$_i$ = (G$_{iM1}$/G$_{iM0}$)/(G$_{iR1}$/G$_{iR0}$). G, $i$, M, R, 0, and 1 denote total growth, genotype, YNB, SC, 0 nM β-estradiol, and 1 nM β-estradiol, respectively (Aux scores for all YETI-NE genes are shown in Dataset EV5B). Z$_3$pr-controlled auxotrophs should see a more dramatic growth improvement when β-estradiol is added to YNB than to SC, and thus, these strains should have large Aux scores. Strains that have reduced growth on YNB medium in the presence of β-estradiol should have small Aux scores.

When Aux scores were ranked from large to small values, as expected, most strains showed no difference in behavior in the presence of β-estradiol on SC vs. YNB medium (Aux scores of 1, Fig 4). However, clear outliers emerged, and thirty-eight of the 42-member metabolic auxotroph panel had Aux scores greater than 1, indicating proper regulation by the ZEV promoter (the 24 top hits from the metabolic auxotroph panel are labeled in blue in Fig 4). Some strains with high Aux scores were not known auxotrophs, and the list includes strains with ZEV alleles driving a variety of metabolic enzymes, as well as several components of the SWI/SNF chromatin modifying complex (the top 17 hits that are not part of the metabolic auxotroph panel are labeled in gray in Fig 4). We validated the growth profile of a strain containing a Z$_3$pr allele of *SHM2*, which encodes a cytosolic serine hydroxymethyltransferase (Appendix Fig S5A). A series of growth experiments revealed that deletion of *SHM2* resulted in adenine auxotrophy, explaining why the Z$_3$pr-*SHM2* strain required β-estradiol to grow on YNB, but not in adenine-containing SC medium (Appendix Fig S5B and C). These examples illustrate that the subset of YETI-NE strains with phenotypes on YNB medium show β-estradiol-dependent growth characteristics reflective of their functional roles.

## Identification of genes that impair growth when overexpressed on plates

The Z$_3$EV allele collection enables tunable regulation of gene expression and a systematic survey of the phenotypic consequences of

gene overexpression. Aggregating plate-based growth data from the YETI-E and YETI-NE panels, we observed that ~ 17% (987/5671) of strains tested (specifically, 301 YETI-E stains and 686 YETI-NE strains) had reduced growth on SC medium at 100 nM β-estradiol as compared to growth at 0 nM β-estradiol (Dataset EV6A, Appendix Fig S6A–D, see Materials and Methods). Among these strains, we saw a direct correlation between increased β-estradiol concentration and the severity of the growth defect. Previous studies of gene overexpression toxicity have explored various mechanisms of sensitivity to protein overproduction, including the balance hypothesis, which posits that deviations in the stoichiometry of protein complex members affect the complex's overall function (Papp *et al*, 2003). While conflicting results have emerged from different gene overexpression studies (Sopko *et al*, 2006; Makanae *et al*, 2013; Moriya, 2015), the 987 toxic genes we identified show no enrichment ($\chi^2$ test, $P = 0.6$) for members of protein complexes (data from Baryshnikova *et al*, 2010; Benschop *et al*, 2010), in agreement with the results of Sopko *et al* (2006). Our list of toxic genes was enriched for genes with annotated roles in regulation of the mitotic cell cycle, DNA replication, microtubule cytoskeleton organization, and chromosome segregation, all bioprocesses related to cell growth that have been reported previously in studies of overexpression toxicity (Dataset EV6B, Gelperin *et al*, 2005; Sopko *et al*, 2006; Douglas *et al*, 2012; Makanae *et al*, 2013). We also found enrichment of several GO function and component terms: transcription factors, nucleocytoplasmic carrier, SUMO transferase activity, and P-body formation.

We examined whether physical properties of proteins were associated with overexpression toxicity. Intrinsically disordered regions (IDRs) tend to make promiscuous molecular interactions when their concentration is increased and have been associated with dosage sensitivity (Vavouri *et al*, 2009). We found that genes whose proteins have a high percentage of predicted IDRs were more toxic for non-essential, but not essential, genes (Appendix Fig S7A and B, Dataset EV6C). Moreover, the toxic gene distribution was significantly skewed toward larger proteins for non-essential genes (KS test, $P$-value $< 2 \times 10^{-14}$, Appendix Fig S7C) but not for essential genes (KS test, $P = 0.38$, Appendix Fig S7D, Dataset EV6C).

Finally, we compared our data with two genome-wide tests of overexpression toxicity that used either the barFLEX (Douglas *et al*, 2012) or the GST-tagged (Sopko *et al*, 2006) collection, both of which employ a GAL-inducible promoter for gene overexpression (Fig EV3A). The GST-tagged collection contains strains expressing ORFs with N-terminal GST tags on high copy plasmids, while the barFLEX collection contains strains expressing untagged ORFs on a low-copy plasmid. To compare screens, we examined only those strains tested in all three studies, which included 608 Z$_3$EV-toxic strains, providing a significant overlap with genes identified in the previous whole-genome overexpression studies ($\chi^2$ test, $P = 1.87 \times 10^{-19}$, Fig EV3A, Dataset EV6D). Overall, we observed a 50% overlap of toxic genes between the YETI collection and barFLEX collection and 30% overlap between YETI and GST-N strains. The larger overlap between barFLEX and YETI may reflect the fact that both collections involve untagged genes. In total, 346 out of 608 of Z$_3$EV-toxic genes were only identified as toxic with the YETI collection (Dataset EV6D). Finally, 27% of universally toxic genes (toxic in YETI, barFLEX, and GST) and 52% of Z$_3$EV-specific toxic genes *only* showed growth impairment at the highest tested

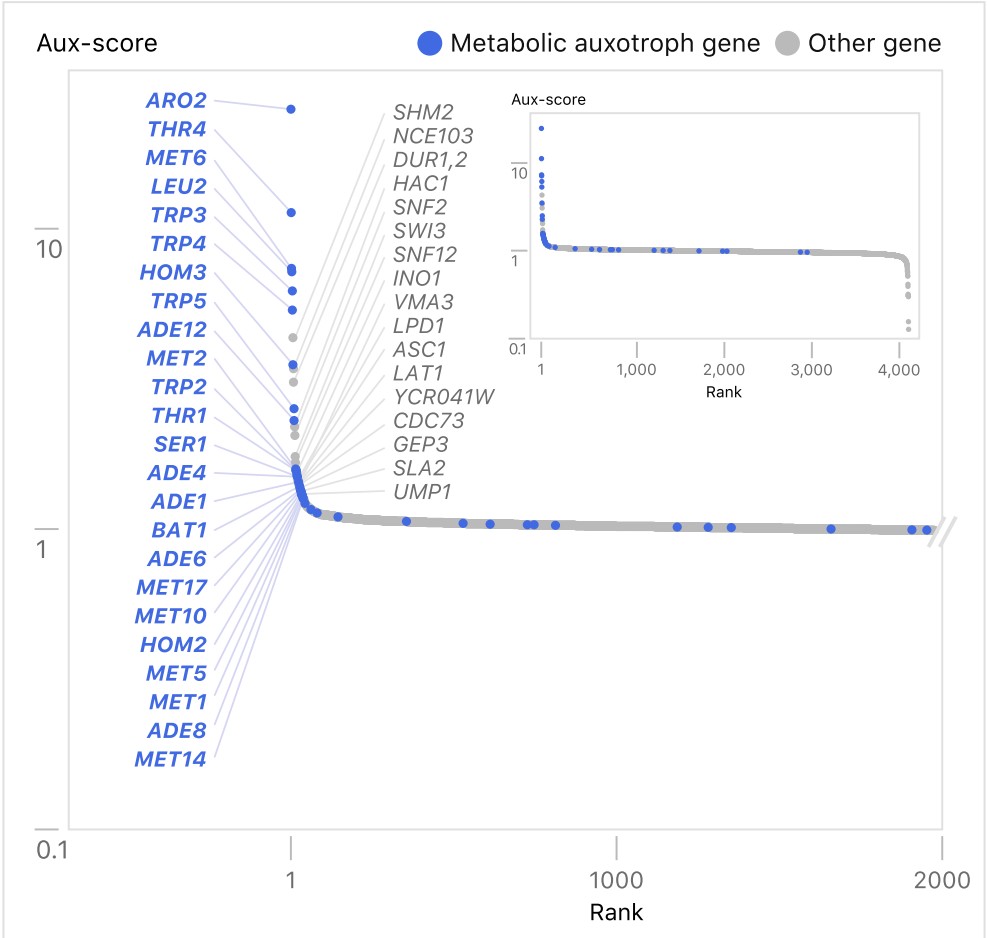

**Figure 4. Comparing β-estradiol-dependent growth of strains expressing YETI-NE genes on YNB and SC.**

Distribution of Aux scores of YETI-NE haploid strains. The *y*-axis shows the Aux score value (see main text and Materials and Methods) measured for each YETI-NE strain displayed on the *x*-axis. YETI-NE strains are shown as gray dots with metabolic auxotroph panel members labeled in blue. The identities of the 41 YETI-NE strains with the highest Aux scores are listed. The inset shows the distribution pattern for all strains tested.

β-estradiol concentration (100 nM, Fig EV3B, Dataset EV6E). In other words, a smaller proportion of universally toxic genes *require* the highest level of inducer to show growth impairment as compared with $Z_3EV$-specific toxic genes. Our ability to detect more growth defects with the $Z_3EV$ collection may be due to a combination of more precise expression control (a titratable promoter and an integrated test gene), as well as use of time-lapse quantification of growth.

**Application #1: Pooled BAR-seq screens**

One of the design features of the YETI collection is the inclusion of unique barcodes marking each $Z_3EV$ promoter allele, enabling pooled growth assays, which use a barcode-based sequencing read-out to assess competitive growth (BAR-seq) (Smith *et al*, 2009; Robinson *et al*, 2014) (Fig 5A). Competitive growth assays are extremely useful for quantifying growth phenotypes associated with mutation of non-essential genes. To test the YETI-NE collection for genes sensitive to under- and overexpression in liquid culture, we inoculated pools into YNB or SC medium with either 0

or 100 nM β-estradiol, under competitive growth conditions, for 48 h in YNB or 36 h in SC. Cultures were re-diluted at successive timepoints to maintain exponential growth and the SC timepoints were chosen such that the number of cell doublings would be similar to those achieved in YNB-grown cultures. At each timepoint, cells were first harvested and then the barcodes were PCR-amplified and quantified via next-generation sequencing. The resulting data were normalized to data collected at time 0, then hierarchically clustered to look for general trends (Dataset EV7A). This analysis revealed 11 clusters of strains which shared patterns of growth depletion dependent on the condition (YNB or SC) and/or the presence of β-estradiol (labeled in Fig 5B). To identify potential biological explanations for the observed growth profiles, we performed functional enrichment analysis on the strain clusters. One cluster, Cluster YNB, was composed of strains depleted in 0 nM β-estradiol in YNB only. Reduced growth in YNB and not SC is suggestive of auxotrophy. Indeed, we saw strong GO enrichment for biosynthetic processes, including amino acid and organic acid biosynthesis, and the average Aux score for "Cluster YNB" genes was a very high (3.18, Dataset EV7B).

Two other clusters were composed of strains that grew more poorly in the absence of β-estradiol, either only in SC medium (Cluster SC) or in both SC and YNB (Cluster "YNB or SC"). We predicted that YETI-NE strains that became depleted over time in 0 nM β-estradiol, but not when their expression was induced, would encode proteins whose expression is important for growth. Consistent with this, both clusters were enriched for genes involved in cytosolic translation (13.5-fold enrichment for Cluster "YNB or SC", $P = 2 \times 10^{-34}$ and 9.9-fold for Cluster SC, $P = 5.9 \times 10^{-8}$) and ribosome biogenesis (10.9-fold for Cluster "YNB or SC", $P = 1.8 \times 10^{-28}$ and 8.3-fold for Cluster SC, $P = 4.8 \times 10^{-7}$), processes that are important for growth. The average single-mutant fitness was 0.65 for genes in Cluster YNB and SC and 0.72 for Cluster SC (Mattiazzi Usaj *et al*, 2020, Dataset EV7B). We also found that genes in these clusters had higher native expression levels, on average, than all YETI-NE genes (Dataset EV7B), consistent with their β-estradiol-dependent growth.

Changes in fitness in culture and on plates were positively and significantly correlated (Spearman correlation = 0.29 at 100 nM β-estradiol, *P*-value $< 2.2 \times 10^{-16}$) (Appendix Fig S8). Additionally, genes identified as toxic upon overexpression on plates showed the strongest agreement with clusters composed of strains that were depleted from the pooled culture in a β-estradiol concentration-dependent manner (Clusters "+e 1a", "+e 1b", and "+e 2"; 96%, 94%, and 72% overlap, respectively; Dataset EV7B). Because we see quantitative agreement between data collected on plates and with BAR-seq, the ease of BAR-seq assays provides a robust paradigm by which the YETI collection can be used for screens in a highly parallel manner.

## Application #2: Mapping transcriptional and genetic interaction networks with YETI

Systematic analysis of transcriptome profiles in yeast strains whose growth is impaired by TF overexpression has been used to discover TF target genes (Chua *et al*, 2006). To explore this approach with the YETI collection, we chose *ROF1*, which encodes a putative transcription factor containing a WOPR DNA-binding domain (Lohse *et al*, 2014; Cromie *et al*, 2017). *ROF1* induction with β-estradiol inhibits growth (Cluster "+e 1b", Fig 5) both on plates (Dataset EV6A) and in BAR-seq (Dataset EV7A) experiments. Overexpression of *ROF1* prevents "fluffy" colony morphology, a proxy for biofilm formation, in the F45 background; hence its name, Regulator of Fluffy (Cromie *et al*, 2017). Deletion of *MIT1*, the paralog of *ROF1*, results in a pseudohyphal growth phenotype in the Σ1278 genetic background, whereas a *rof1Δ* mutant has no phenotype in this assay (Cain *et al*, 2012).

The growth phenotype associated with *ROF1* overexpression in the Z₃pr-*ROF1* strain provided an opportunity to explore the *ROF1* regulon and to illuminate *ROF1* function. We grew the Z₃pr-*ROF1* strain to steady state in a phosphate-limited chemostat, induced *ROF1* with 1 μM β-estradiol, and tracked genome-wide gene expression changes over 90 min, as described previously (McIsaac *et al*, 2012; Hackett *et al*, 2020). The *ROF1*-YETI strain has less *ROF1* transcript than a WT strain does at *t* = 0 min. Upon induction with β-estradiol, in duplicate experiments, we observed a subset of genes whose expression was rapidly reduced upon *ROF1* overexpression, suggesting Rof1 acts as a transcriptional repressor (Fig 6, Dataset EV8A). Using HOMER (Hypergeometric Optimization of *Motif* EnRichment; Duttke *et al*, 2019), we found that the promoters of genes repressed by Rof1 are most strongly enriched for a WOPR-domain-like DNA-binding sequence, [A/T]TTAAACTTT (*P*-value ~ $10^{-11}$). The Rof1-repressed genes were enriched for several functional processes, including water transport, siderophore transmembrane transport, and repressing transcription factor binding activity (Dataset EV8B). Several TF genes were repressed by Rof1, including *MOT3, CUP9, PHD1, YAP6, NRG1, GAT2, TBS1, ACA1, GCN4, CIN5*, and a gene encoding Rof1-paralog, *MIT1*. Our initial analysis of *ROF1* indicates that it encodes a *bona fide* TF that is highly interconnected with other TFs and illustrates the usefulness of ZEV promoter strains for exploring TF function.

Synthetic dosage lethal (SDL) interactions, which occur when overexpression of a gene causes a more severe fitness defect when a second gene is mutated, can reveal information about pathways and bioprocesses affected by the overexpression. YETI strains carry SGA markers, enabling introduction of Z₃EV induction alleles into appropriately marked arrays of yeast mutant strains. We used this feature to further explore the genetic basis of the fitness defect caused by *ROF1* overexpression. We crossed a query strain containing Z₃pr-*ROF1* with the deletion collection and the collection of temperature-sensitive alleles of essential genes in the absence of β-estradiol, and then we induced *ROF1* overexpression with 1 μM β-estradiol at the final SGA selection step. We discovered 264 unique genes that had SDL interactions with *ROF1* (Dataset EV8C), with the strongest interactions involving members of the cell wall integrity (CWI) pathway, *SLT2, BCK1*, and *PKC1*, and two genes with roles in cell wall biosynthesis, *GFA1* and *QRI1*, consistent with a role for *ROF1* in surviving cell wall stress (Jiménez-Gutiérrez *et al*, 2020). *ROF1* also showed SDL interactions with numerous genes involved in chromatin remodeling and general transcription, including multiple members of the Swr1 complex, Ino80 complex, Rpd3L complex, NuA4 complex, and COMPASS, which presumably reflects its role as a transcriptional repressor and a regulator of other TFs. Comparison of the *ROF1* overexpression microarray profile to microarray profiles of strains deleted

**Figure 5. BAR-seq of YETI-NE haploid strains.**

A   Diagram illustrating the pooled culture growth and harvesting strategy for BAR-seq experiments with YETI-NE strains. The labels in the diagram are also used to indicate the relevant growth conditions in the corresponding clustergram (part B).

B   BAR-seq analysis of YETI-NE pools in SC and YNB medium in the absence or presence of 100 nM β-estradiol. Data were hierarchically clustered and 11 clusters labeled based on conditions in which strains exhibited reduced growth: Cluster YNB +e (showing depletion in 100 nM β-estradiol in YNB only), Cluster YNB (depletion in 0 nM β-estradiol in YNB only), Clusters All 1 and All 2 (time-dependent depletion in all conditions), Cluster YNB or SC (depletion in either YNB or SC at 0 nM β-estradiol), Clusters "+e 1a" and "+e 1b" (fast depletion in YNB [12 h] and SC [9 h] at 100 nM β-estradiol), Cluster "+e 2" (depletion by 24 h in YNB and 18 h in SC at 100 nM β-estradiol), Clusters "SC+e 1" and "SC+e 2" (depletion in SC only at 100 nM β-estradiol), Cluster SC (depletion in SC only at 0 nM β-estradiol). Experiments were performed in biological triplicate.

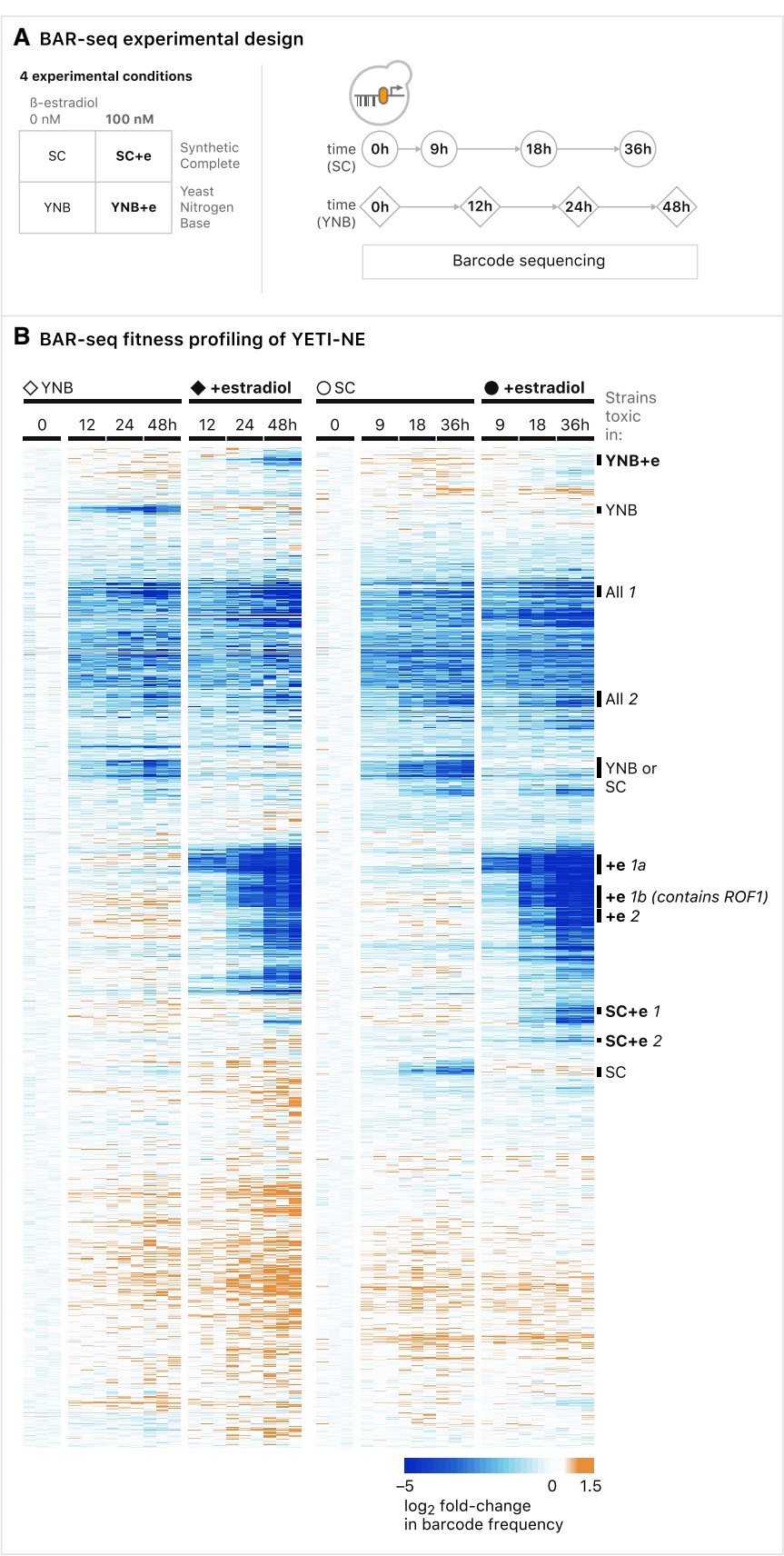

**Figure 5.**

for chromatin regulators did not reveal any obvious pattern of co-regulation that might provide a mechanism for the SDL interactions (Lenstra *et al*, 2011). We speculate that overexpressing *ROF1* causes a stress response making cells dependent both on the CWI pathway and on multiple transcription pathways for normal growth.

### Reversibility of the Z₃EV promoter alleles

Our experiments with a Z₃EV-GFP allele showed that the Z₃EV promoter could be both induced by the addition of β-estradiol and repressed upon its removal (Fig 1C). This is an important feature that could prove particularly useful for the study of essential genes. We examined whether expression of the Z₃EV promoter was more generally reversible by assaying the growth of strains carrying Z₃EV promoter alleles of essential genes. We chose 24 Z₃EV strains whose growth was β-estradiol-dependent, sporulated the Z₃EV diploids, and then selected haploids containing the Z₃EV system in the presence of 10 nM β-estradiol, which allowed good growth for all the strains. We then transferred the strains to the same medium containing 0 nM β-estradiol or 10 nM β-estradiol, incubated for two days, and then repeated the transfer and growth (Dataset EV9). Three of 24 strains failed to grow on 0 nM β-estradiol, showing strong reversibility of the Z₃EV promoter allele, and another five strains had Z₃EV promoter alleles that showed partial reversibility in this assay. However, 16 of the 24 (67%) Z₃EV strains grew in the absence of β-estradiol, even though their initial growth had been β-estradiol-dependent (Appendix Fig S9). There was no difference in native expression between the 16 genes that lacked reversibility and the eight genes that showed either full or partial reversibility (*P*-value = 0.35, two-sided *t*-test). It is possible that some essential genes under the control of Z₃EV continue to be expressed in the absence of β-estradiol for numerous cell doublings. We previously found that changes in DNA accessibility using ATAC-seq were correlated with future changes in RNA expression (Hendrickson *et al*, 2018a). Mechanistically, increased DNA accessibility at Z₃pr following β-estradiol addition may not completely reverse upon β-estradiol removal, resulting in different levels of phenotypic reversibility depending on the target gene.

We also examined reversibility of overexpression toxicity (see Materials and Methods for details). We chose 19 strains from the YETI-NE panel carrying Z₃EV alleles that caused at least a ~ 50% growth defect in the presence of 100 nM β-estradiol (Appendix Fig S10A). These strains were first pinned onto SC lacking β-estradiol, and then, colonies were transferred to fresh YNB plates either with or without β-estradiol. After ~ 24 h of growth, colonies were transferred again to YNB plates with or without β-estradiol to initiate a 120-h time-lapse growth assay. Of the 19 toxic Z₃EV promoter alleles tested, 13 (68%) were fully or partially reversible. For example, the Z₃EV-*TIP41* allele showed a strong reversibility phenotype (Appendix Fig S10B); when pinned from 10 to 0 nM β-estradiol, growth is largely restored to normal levels (Appendix Fig S10C). In contrast, the Z₃EV-*SGS1* allele caused toxicity that was not reversible (Appendix Fig S10D). Although we have only examined a subset of Z₃EV promoter alleles for reversibility, our analyses suggest that overexpression toxicity phenotypes associated with Z₃EV promoter allele expression are more easily reversed than growth defects that are caused by removing β-estradiol from the growth medium.

### Z₃EB42: a reversible artificial transcription system for lowly expressed genes

Our characterization of the Z₃EV system revealed limitations, including the growth of many strains expressing Z₃EV promoter alleles of genes with low endogenous expression even in the absence of inducer (Fig 2). Previous work has shown that changing the number of Zif268 binding sites in the β-estradiol-dependent promoter, and/or modifying the activation domain of an artificial transcription factor (ATF), can expand the range of possible gene expression levels (McIsaac *et al*, 2014; Ottoz *et al*, 2014). Therefore, we re-engineered Z₃EV as well as its target promoter, Z₃pr, with the goal of designing a system with reduced "leakiness" of gene expression in the absence of inducer. Specifically, we engineered promoters with different numbers of Zif268 binding sites (6 or 2) (Figs 7 and EV4). We also employed the use of yeast regulatory elements to attempt to further reduce the strength of Z₃pr (Agmon *et al*, 2017). Between position −158 and −146 of the *CAR1* gene is an upstream repression sequence (URS1), required for *CAR1* repression (Luche *et al*, 1990). Ume6, a DNA-binding protein, binds to the URS1 sequence and recruits the Sin3-Rpd3 complex for transcriptional repression (Kadosh & Struhl, 1997). Ume6 represses sporulation-specific genes during mitotic growth (Strich *et al*, 1994) and genes involved in arginine catabolism when readily catabolized nitrogen sources are available in the environment (Messenguy *et al*, 2000) so we anticipated that it would have repression activity under our standard conditions. Thus, we appended a region extending from nucleotides −198 to −1 of *CAR1* to the 3'-end of Z₃pr (6× or 2× Zif268 binding sequences). Finally, we created a new ATF, Z₃EB42, with a B42 transcriptional activation domain in lieu of VP16. B42, a short unstructured acidic peptide encoded by an *E. coli* genomic DNA fragment, has weaker activity than VP16 (Ruden *et al*, 1991; Ottoz *et al*, 2014).

We constructed eight combinations of ATFs and Z₃ promoters and quantified dose-response curves across a range of β-estradiol concentrations using an mNeonGreen reporter (Figs 7 and EV4). As predicted, URS1 had a strong effect on mNeonGreen production, resulting in a 20-fold decrease in production for Z₃EV and a 14-fold decrease in production for Z₃EB42 for 2× Zif268 binding sites in high β-estradiol (Fig 7). Additionally, at saturating levels of inducer, we saw that the use of Z₃EB42 resulted in ~ 2-fold less fluorescent signal than Z₃EV for each promoter. Consistent with previous work, reducing the number of Zif268 binding sites in Z₃pr reduced fluorescent signal (McIsaac *et al*, 2014) (Fig 7). The weakest promoter in our panel was Z₃(2× Zif268 binding sites + URS1)pr, which we refer to as Z₃pr-Zif2-URS1. The combination of Z₃EB42 and Z₃pr-Zif2-URS1 resulted in the least leaky expression in the absence of inducer, as well as the lowest mNeonGreen signal. Finally, we assessed the reversibility of each ATF-promoter pair, and we found that URS1 made an important contribution to reversibility: for both Z₃EV and Z₃EB42, only promoters containing URS1 were able to reverse expression to background levels in < 24 h (Appendix Fig S11). Z₃EB42 and Z₃pr-Zif2-URS1 were the most reversible ATF-promoter pair. We now refer to the combination of the ATF, Z₃EB42, and the synthetic promoter, Z₃pr-Zif2-URS1, as the Z₃EB42 system (Fig 8A).

We compared Z₃EV and Z₃EB42 function on promoter replacement alleles of a panel of essential genes: three that enabled growth

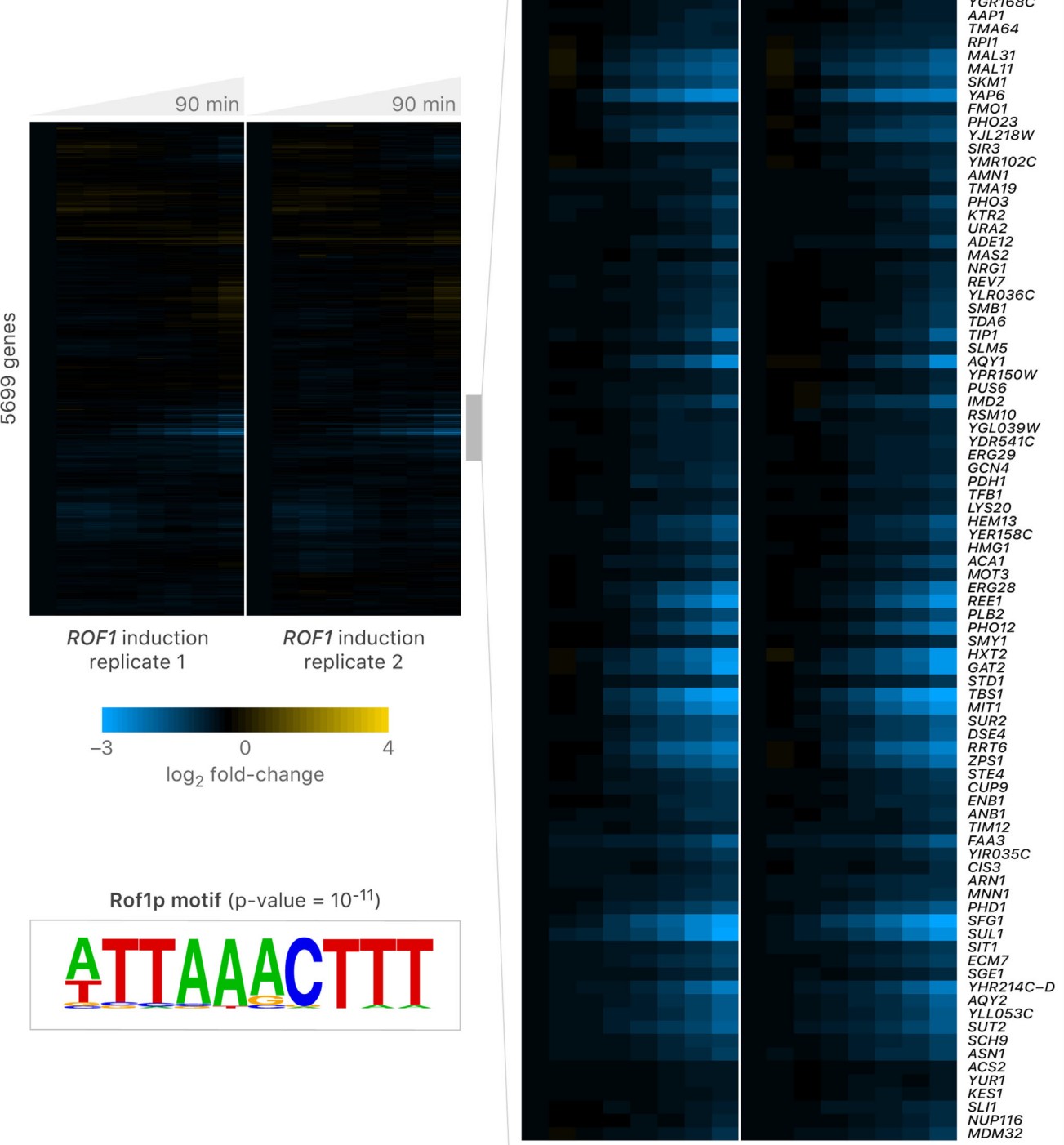

**Figure 6. Rof1 is a transcriptional repressor.**

Microarray expression data resulting from induction of Z₃pr-*ROF1* over a 90-min time-course. The Z₃pr-*ROF1* strain was grown in the presence of 1 μM β-estradiol in a phosphate-limited chemostat, and samples were harvested at the indicated times. Two independent time courses are shown. Gene expression data were clustered using a Pearson correlation distance metric. One regulon of Rof1-repressed genes was identified, which is highlighted on the right. Motif enrichment was calculated using the cumulative binomial distribution in the HOMER software suite (Heinz *et al*, 2010).

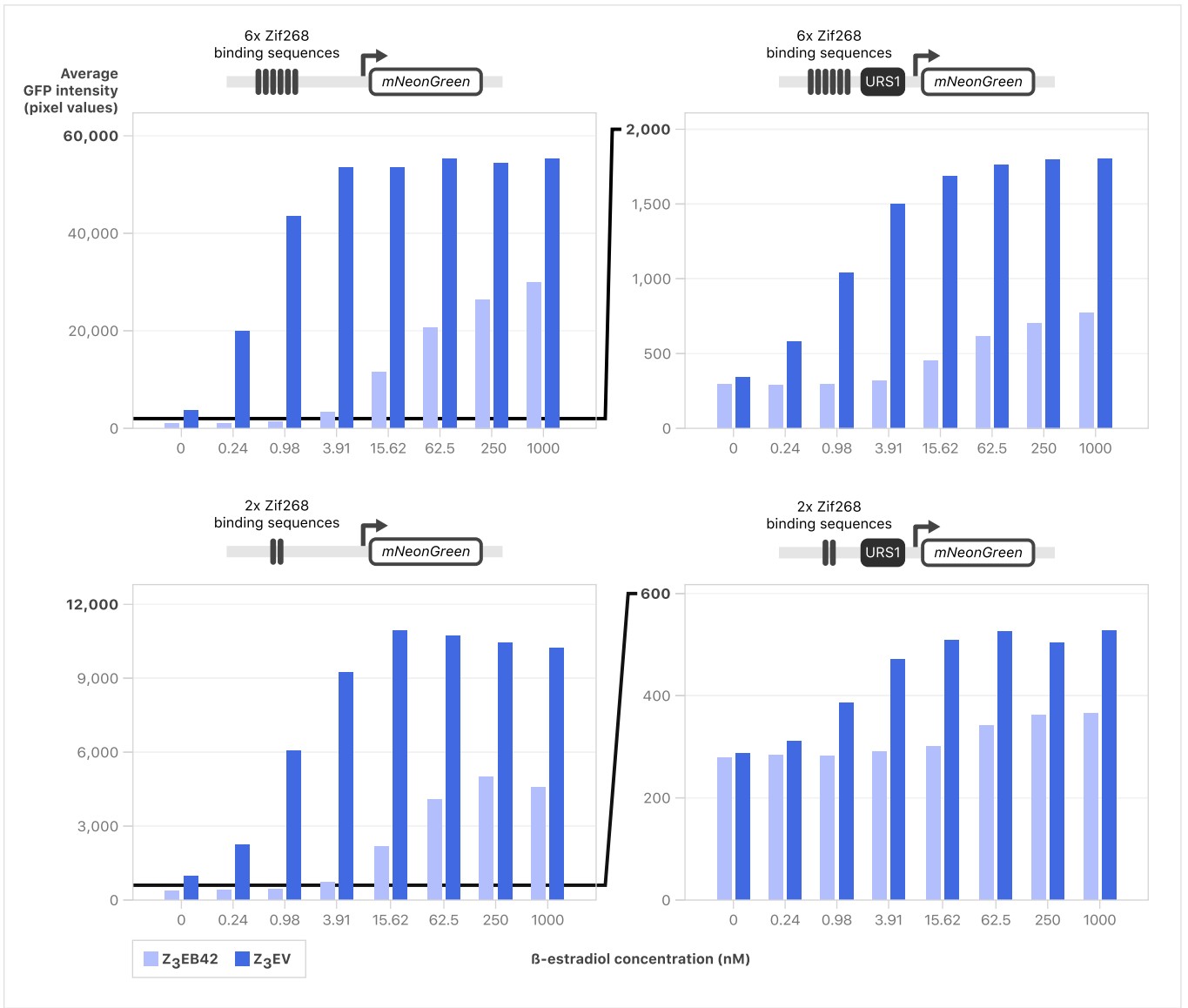

**Figure 7. Number of binding sites, URS1 presence, and ATF choice all affect mNeonGreen reporter induction.**

Strains with all combinations of ATFs and promoters driving mNeonGreen were grown to early log phase in YNB and mNeonGreen was induced with various levels of β-estradiol. Strains were imaged every hour using a Phenix automated confocal microscope and average fluorescence/cell was calculated using Harmony software. Average of two replicates is shown at $t$ = 8 h following induction with > 300 cells quantified per dose. Full time series are shown in Fig EV4.

without β-estradiol (*RAD53*, *CIA2*, and *IPL2*), one that allowed growth without inducer but was toxic when overexpressed (*PBR1*), and one that gave β-estradiol-dependent growth (*HYP2*) with the $Z_3EV$ system. Strains with the $Z_3EV$ system driving *RAD53*, *CIA2*, or *IPL2* grew in the absence of β-estradiol (Fig 8B left). Strains with these genes under the control of $Z_3EB42$ were not able to grow in 0 or 10 nM β-estradiol, indicating that any leaky expression was repressed with the stringent system (Fig 8B right). *PBR1* was leaky and had a growth defect on 1,000 nM β-estradiol when regulated by the $Z_3EV$ system (Fig 8C left). In contrast, a strain with $Z_3EB42$ system driving *PBR1* expression did not grow in the absence of β-estradiol but grew in 1,000 nM β-estradiol (Fig 8C right). The growth defect phenotype caused by *PBR1* overexpression in

1,000 nM β-estradiol with the $Z_3EV$ system was not detected with $Z_3EB42$. Finally, we tested a $Z_3EB42$-inducible allele that behaved well in the YETI allele collection, with no growth in the absence of β-estradiol and normal growth in the presence of 10 nM or higher β-estradiol (Fig 8D left). The $Z_3EB42$-regulated *HYP2* did not support growth even at 1,000 nM β-estradiol (Fig 8D right). These results indicate that, for some essential genes, $Z_3EB42$ leads to levels of gene expression low enough that growth is β-estradiol-dependent, but for others, it may not enable enough expression to support growth and may not cause phenotypes that depend on overexpression.

Since we found that the $Z_3EB42$ system conferred β-estradiol-dependence for some essential genes that were not regulated with

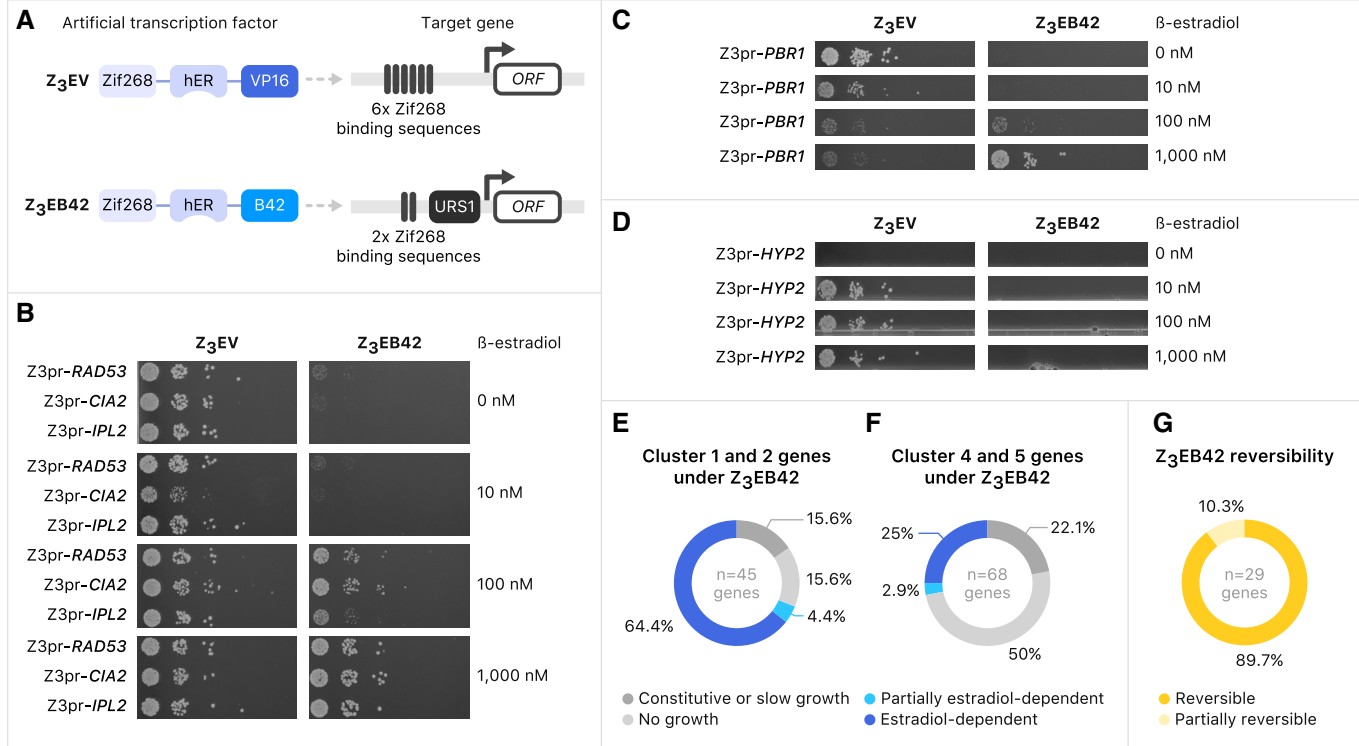

**Figure 8. Engineered gene expression system for more stringent regulation.**

A  Diagrams of $Z_3EV$ artificial transcription factor with $Z_3pr$ promoter and $Z_3EB42$ artificial transcription factor with $Z_3pr$-Zif2-URS1 promoter.

B–D  Serial spot dilutions onto medium containing various concentrations of β-estradiol. Diploid strains containing the indicated regulatory systems (see (A)) controlling the target genes were incubated in sporulation medium for 5 days. Serial dilutions of cells were plated on SD-his-ura-arg-lys with canavanine, thialysine, and ClonNAT with indicated concentrations of β-estradiol to obtain haploid cell growth.

E  Distribution of growth phenotypes for strains carrying $Z_3EB42$ promoter alleles for 45 essential genes from Cluster 1 and 2 (see Fig 2).

F  Distribution of growth phenotypes for strains carrying $Z_3EB42$ promoter alleles for 68 essential genes from Cluster 4 and 5 (see Fig 2).

G  Distribution of $Z_3EB42$ promoter reversibility phenotype for 29 tested essential genes.

Data information: The color key for interpreting the figure is below panels (E–G).

$Z_3EV$, we expanded our survey of $Z_3EB42$ system performance to include a larger set of essential genes that had given either β-estradiol-dependent (Clusters 4 and 5) or β-estradiol-independent (Clusters 1 and 2) growth with the $Z_3EV$ system. Of 45 strains with genes that were β-estradiol-independent, 7 (15.6%) gave constitutive growth, 29 (64.4%) were β-estradiol-dependent, 2 (4.4%) were partially β-estradiol-dependent and 7 (15.6%) gave no growth when controlled by $Z_3EB42$ instead of $Z_3EV$ (Fig 8E, Dataset EV10). Of 68 strains with genes that were β-estradiol-dependent with $Z_3EV$, 15 (22%) were constitutive, 17 (25%) were β-estradiol-dependent, 2 (3%) were partially β-estradiol-dependent, and 34 (50%) were not able to grow at any β-estradiol concentration when controlled by $Z_3EB42$ (Fig 8F, Dataset EV10). Because the genes from Clusters 4 and 5 tend to have higher native transcript levels, these data suggest that the $Z_3EB42$ system does not drive sufficient gene expression to support growth for a significant fraction of highly expressed essential genes. Finally, we asked if the lower-expression $Z_3EB42$ system could confer reversible growth. Of 29 genes from Cluster 1 and Cluster 2 that had β-estradiol-dependent growth with $Z_3EB42$, we found 26 (90%) were reversible and 3 (10%) were partially reversible (Fig 8G). These data demonstrate that the $Z_3EB42$ system confers

high reversibility of essentiality and provides a useful tool for investigating the loss-of-function of Cluster 1 and 2 genes.

We conclude that the $Z_3EV$ system can achieve conditional growth for many essential genes and that the $Z_3EB42$ system tends to enable conditional (and reversible) growth in cases where $Z_3EV$ cannot. Thus, when growth is the readout, the best choice of expression system depends on the individual properties of a specific gene.

## Discussion

We constructed and characterized YETI, a genome-scale collection of $Z_3EV$ inducible alleles of yeast genes. Previously, we established that combining inducible expression with transcriptome-wide time series measurements is an effective strategy for elucidating gene regulatory networks. The ability to switch a gene on and measure dynamically how every other gene responds has allowed us to identify causal regulatory interactions, both direct and indirect, and observe instances of feedback control that were previously inaccessible (McIsaac *et al*, 2012; Hackett *et al*, 2020). Additionally, all engineered components are directly integrated in the genome (i.e.,

no plasmids), a feature that is especially useful for achieving homogeneous expression of individual target genes within a population of cells. By creating the YETI collection, we are expanding the uses of this system from studies of individual genes to nearly all genes in the yeast genome. To characterize the YETI collection and provide information to guide its use by the community, we have performed quantitative growth-based phenotyping and carried out pooled BAR-seq screens. To demonstrate the utility of this YETI collection, we mapped the regulon of a putative TF and identified its SDL interactions. Collectively, these experiments validated the quality of the library, revealed new biology, illuminated an important though unappreciated design principle of synthetic gene control, and motivated the development of the $Z_3EB42$ system for applications requiring more refined regulation.

Because our YETI strains were constructed first as diploids, we harnessed the power of yeast genetics to sporulate and select haploid strains with one copy of $Z_3pr$-controlled essential genes at differing levels of inducer. This unique experimental design revealed that ~ 40% of $Z_3pr$-controlled essential genes had enough transcript to facilitate growth in the absence of inducer, even though $Z_3pr$ is extremely weak. Many genes in bacteria, yeast, and humans are lowly expressed with only a handful of transcripts per cell (Raj & van Oudenaarden, 2008). Making a strong inducible system that is also completely "off" in the absence of β-estradiol is an engineering challenge. Indeed, the cost of achieving strong inducible expression is often some basal level of leakiness, which can be problematic for growth-based screens. To determine the extent of this problem, in yeast and *E. coli*, we took advantage of a ground-truth set of essential genes.

Unlike yeast and *E. coli*, we have yet to define a clear-cut ground-truth dataset of essential genes in human cells. Many human essential genes have been identified in a variety of different cell lines from pooled screens, and we find that annotated essential genes are ~ 10×–20× more highly expressed than those defined as non-essential (Fig EV5). In yeast, essential genes are, on average, expressed only ~ 1.4× more highly than non-essential genes. The difference in expression between essential and non-essential genes could be much greater in human cells than it is in yeast, or pooled screens in cell lines may be sampling the most extreme parts of the expression distribution. We favor the latter interpretation and believe within these screens that the loss of highly expressed essential genes is simply easier to detect than the loss of lowly expressed essential genes with currently available technology. This interpretation also implies that CRISPR gene inactivation approaches are more effective at disrupting highly expressed genes than lowly expressed ones. Mechanistically, low levels of chromatin accessibility, which is associated with low levels of gene expression, could prevent both Cas9 and dCas9 from accessing specific gene targets. Focusing on yeast and *E. coli*, we found that generating low expression of already lowly expressed essential genes, using multiple approaches, including $Z_3EV$, Tet-off, DAmP, TS-degron, and CRISPRi, is problematic for growth-based screens. In yeast, recently improved Tet-based systems with low expression variation and reduced leakiness may be useful for some applications, including expressing lowly expressed genes, as well as multiplexed experiments with the YETI collection (Roney *et al*, 2016; preprint: Azizoğlu *et al*, 2020). Here, we developed $Z_3EB42$ to specifically enable conditional and reversible growth with lowly expressed essential genes.

The YETI collection provides new opportunities to elucidate regulatory interactions and instances of feedback control on a genome scale. We anticipate that the barcode feature will enable this collection to be used for highly parallel pooled screens (Robinson *et al*, 2014). Indeed, our collection provides an opportunity to perform pooled screens without plasmids and at many different levels of expression. Furthermore, the YETI collection provides a new and general system for functional analysis. Of particular interest to us is that replicative aging in yeast is a model for studying the aging process in eukaryotes. For example, it should be possible to combine the YETI collection with Miniature-chemostat Aging Devices (MADs) (Hendrickson *et al*, 2018b) to explore how molecular networks change with replicative age. MADs give us the ability to obtain replicatively aged "mother cells" that are rare in standard cultures. Two experimental paradigms are possible. First, combined with YETI, we anticipate that we may be able to explore how titrating individual genes can alter or reprogram the aging process itself. The relationship between the expression level of a specific allele and a marker that reports on an age-related change (i.e., vacuolar fragmentation (Lee *et al*, 2012)) could be investigated one strain at a time, or potentially through a pooled screening approach. Second, we will be able to explore if and when there are age-dependent changes in molecular networks by performing experiments similar to that shown in Fig 6 at multiple ages. We also believe that there are opportunities in metabolic engineering and industrial biotechnology applications for titrating target genes to increase the yields of useful products. To make the YETI collection compatible with perturb-Seq-like approaches for monitoring regulatory networks at the single-cell level, barcodes encoded in RNA could be introduced (Jackson *et al*, 2020). Finally, each strain in this collection contains the SGA markers, which means future strain engineering can be done using automated approaches. The combination of YETI with fluorescent markers will enable phenotypic screens that can be performed in pooled or arrayed formats.

# Materials and Methods

### Strain and plasmid construction

The yeast strains and plasmids used in this study are listed in the Appendix. All media recipes are provided in the Appendix. To construct the parental diploid yeast strain Y14789, expressing the $Z_3$ binding domain-hER-VP16 transcription factor ($Z_3EV$) and carrying SGA markers, a DNA fragment containing a *natMX*-marked $Z_3EV$ fragment was PCR-amplified from strain DBY12394 using primers that contained sequences that enabled integration into the genome downstream of the repaired *HAP1* locus in the strain RCY1972 (gift from Dr. Amy Caudy). The strain was then crossed to Y7092 to create a diploid strain, and Y14851 was isolated by tetrad dissection. Subsequently, Y14851 was crossed to Y14537 to generate diploid strain Y14789, which was used for construction of the β-estradiol-inducible allele strain collection. To construct the source plasmid for $Z_3EVpr$ (p5820), the *kanMX* marker in plasmid pRB3564 [13] was replaced with the *S. cerevisiae URA3* gene. The $Z_3$ promoter ($Z_3pr$) is a derivative of the *GAL1* promoter in which a region containing three canonical *GAL4* binding sites (5'-CGG-$N_{11}$-CCG-3') is replaced with six Zif268 binding sites (5'-GCGTGGGCG-3'). In

creating p5820, we also removed a non-canonical Gal4 binding site from pRB3564. The sequence of the $Z_3EV$ transcription factor coding region was previously described. The plasmids p7418 and p7419 were constructed by modifying p5820.

To construct β-estradiol promoter replacement alleles for each *S. cerevisiae* gene, the $Z_3$ synthetic promoter linked to *URA3* was PCR-amplified from plasmid p5820. The forward primer contained homology to 40 bp upstream of the ATG of each gene with a unique 12 nucleotide barcode. The reverse primer contained 40 bp complementary to the region immediately downstream of the ATG (Dataset EV2A–C). PCR products were transformed into strain Y14789 and transformants were selected on SC-ura+ClonNAT at 30°C. The integration locus was confirmed by colony PCR, as previously described [27]. Candidate clones were streaked onto YPD+ClonNAT plates, and single colonies were isolated and replica-plated onto SC-ura to check the stability of the integrated marker. For non-essential ORFs, we also constructed a *MAT***a** haploid collection, by sporulating the relevant diploid strains for 5–6 days. Following sporulation, single colonies were isolated by streaking the spore mixture onto SGA selection medium SC-his-ura-arg-lys with canavanine, thialysine, and ClonNAT [28], and the plates were incubated at 30°C for 4 days. The integration locus was re-confirmed by colony PCR. The mating types and selectable markers of the haploid strains were also confirmed. To make Y15292, a GFP fragment and the $Z_3$ synthetic promoter were PCR-amplified then integrated into the *ho* locus of Y14851. For testing the stringent system, $Z_3EB42$ was integrated downstream of the repaired *HAP1* locus. To make plasmid p7418, the six Zif268 binding sites of $Z_3EVpr$ (p5820) were replaced with two Zif268 binding sites. One hundred and ninety-seven base pairs of *CAR1* upstream sequence including URS1 were appended to the 3' end of the promoter (Appendix).

The YETI-E and YETI-NE diploid collections have the genotype: *MAT***a**/α   ORF+/[*barcode::URA3::Z3EVpr-ORF*]   [*HAP1::natMX-ACT1pr-Z3EV-ENO2term*]/*HAP1*   *ura3Δ0/ura3Δ0*   [*can1Δ::STE2pr-Sphis5*]/*CAN1 his3Δ1/his3Δ1 lyp1Δ/LYP1*. Following selections, the YETI-E and YETI-NE haploid genotypes are *MAT***a** [*barcode::URA3:: Z3EVpr-ORF*]   [*HAP1::natMX::ACT1pr-Z3EV-ENO2term*]   *ura3Δ0 can1Δ::STE2pr-Sphis5 his3Δ1 lyp1Δ*.

### Assessing expression of GFP using flow cytometry

To assess the dependence of gene expression on the concentration of β-estradiol in the experiments shown in Fig 1, the yeast strain Y15292, carrying a GFP reporter gene driven by $Z_3pr$, was first incubated overnight at 30°C in YNB. The overnight culture was then diluted into fresh medium and grown at 30°C to a density of $6 \times 10^6$ cells/ml. Cells were incubated with β-estradiol for 6 h. To measure the effect of removing β-estradiol on GFP expression, cells were washed in PBS 3 times and then incubated in fresh medium without β-estradiol. At each timepoint, cells were harvested and fixed in 70% ethanol. Subsequently, cells were washed three times with 1 ml PBS and then 5 μg/ml propidium iodide was added. To measure GFP signal in single yeast cells, we used fluorescence-activated cell sorting to record 10,000 events using a BD FACS Aria lllu with a high-throughput sampler (BD Biosciences). We gated for single cells, which were identified by analyzing the signal width of the forward and the side scatter. Dead cells, which were stained with propidium iodide, were excluded from data analysis.

### $Z_3$(2 binding sites) URS1 promoter alleles of essential genes

Growth of essential gene alleles was assessed by sporulating the heterozygous diploid strains on sporulation plates for 5 days and then either spotting (with 10× dilutions) or directly pinning onto SC-his-ura-arg-lys with canavanine, thialysine, and ClonNAT plates in the absence or presence of various concentrations of β-estradiol.

### Preparation of agar plates for growth profiling of YETI-E and YETI-NE strains

All the agar plates were prepared by either Universal Plate Pourer (KREO Technologies Inc.) or manually in Nunc™ OmniTray™ Single-Well Plate (Thermo Fisher Scientific 264728). To minimize noise of colony size, medium volume in automatically poured plate was set to 40 ml, and plates with air bubbles, particles, uneven surfaces, and scratches on the bottom of the plates were discarded. After agar was solidified, plates were flipped (upside down) and dried for 48 h at room temperature to maintain the same moisture level in each plate.

### Pinning and imaging of agar plates

Pinning of yeast strains was done with a BM3-BC pinning robot (S&P Robotics, Inc.) using various pin tools indicated below. Incubation of plates at 30°C and imaging of plates were done using the spImage-A3 instrument (S&P Robotics, Inc.) with manual focus mode of camera setting. To reduce the time lag from the pinning and imaging of plates, less than 32 plates were processed at once, and imaging was started immediately after pinning on final assay plates with β-estradiol. Plate images were generated every hour up to the time point indicated in each experiment (mostly up to 60 h).

### Image processing, quantification of yeast colonies, and growth phenotypes

For most of the reported data, colony sizes were quantified using Platometer, an open-source image-processing Python tool available at https://github.com/baryshnikova-lab/platometer. Briefly, Platometer automatically detects the grid of colonies on an image and estimates the size of each colony as the number of pixels above background after adaptive thresholding. Following quantification, raw colony sizes were normalized for plate effects, positional effects, and competition effects as described previously (Baryshnikova *et al*, 2010). Images were taken every hour for 60 h, and normalized colony sizes across consecutive timepoints were assembled into growth curves. For each colony, the area under the growth curve (AUGC), which integrates the main three aspects of yeast population dynamics (duration of lag phase, exponential growth rate, and carrying capacity), was used as a global estimate of growth efficiency.

### Growth profiling of YETI-E strains

YETI-E strains (1,022 heterozygous diploids) were re-arrayed to randomize positions of each strain on agar plates in 384 format. One row and one column on the edge was filled with wild-type strain (Y15090) to exclude nutritional and spatial advantage for the

strains on the edge. Two sets of the re-arrayed (position-randomized) YETI-E collection were pinned again to generate quadruplicate 1,536 format, and multiple copies were prepared using a 1,536 0.8 mm pin tool. All the pinnings for re-arraying and copying were done on "Z₃EV diploid maintenance plates". The colonies were pinned onto "enriched sporulation plates" for Z₃EV diploid and incubated at room temperature for 5 days. Thereafter, sporulated colonies were pinned onto "Z₃EV essential haploid selection with β-estradiol plates" (see Appendix) with a 1,536 0.5 mm pin tool. 0.5 mm pins are the lower limit in size to deliver small amounts of yeast cells to destination plates to generate a larger dynamic range of growth, and not to puncture agar plates. Two technical replicates were processed for two position-randomized libraries at 12 β-estradiol concentrations (0, 0.01, 0.03, 0.1, 0.3, 1, 3, 10, 30, 100, 300, and 1,000 nM).

### Growth profiling of YETI-NE strains

The non-essential haploid collection containing ~ 4,600 strains was processed in the same manner as the essential collection except that the wild-type strain on the edge was different (Z₃pr-*ho*), and the library came from the "Z3EV haploid maintenance plate" (complete media) and was pinned onto SC or minimal media. Two sets of experiments were performed: pinning from complete medium to complete medium with β-estradiol and pinning from minimal medium to minimal medium with β-estradiol. Doses of β-estradiol were 0, 1, 5, 10, and 100 nM.

### Whole-genome sequencing of strains

Additional QC of the library was done by whole-genome sequencing of a subset of strains to check for correct insertion of Z₃EVpr as well as to look for any potential signs of aneuploidy. Cells were grown overnight in a diluted culture until log phase was reached in the morning. Samples were first processed using the YeaStar Genomic DNA kit (Zymo #D2002) to obtain genomic DNA and prepared for whole-genome sequencing using the Nextera DNA Flex Library prep kit (Illumina #20018705). Samples were sequenced on the NovaSeq 6000. Read depth over all chromosomes was then plotted using Python to detect instances of aneuploidy.

A total of 107 strains were selected to cover both essential and non-essential strains: 27 essential diploid strains and 40 non-essential strains in diploid and haploid. Overall, 95.3% of strains show proper insertion and attachment of the Z₃EVpr. 86.0% of all strains in our panel showed no signs of aneuploidy. Strains in this panel are enriched for haploids involved in pathways that get selected against when undergoing the SGA selection procedure (URA, HIS, LYS, and ARG selections/genes). If we remove those strains from our analysis, then 95.9% of all strains tested showed no signs of aneuploidy in our panel.

### Strain construction statistics

Correct insertion of the URA3::Z₃pr cassette was confirmed by PCR for all strains. Only 114/5804 strains (2%) failed construction, even after multiple attempts, as assessed by absence of PCR product in the confirmation assay. Some of the failed ORFs are close to transposable elements or other repetitive sequences, which could result

in the absence of a specific band in our confirmation PCR (Dataset EV2). We analyzed growth on β-estradiol to further assess whether strains confirmed by PCR had acquired defects (e.g., through PCR or recombination) such that the Z₃pr was unable to induce expression of the gene. Following our initial construction, we identified 90 strains that had little or no growth at all concentrations of β-estradiol. We attempted to re-make 54 of these strains and were able to construct 47 that had growth at some concentration of β-estradiol (Dataset EV3, reconstructed strains), suggesting that most of the 90 that initially had little or no growth in response to β-estradiol were defective for their original construction. We believe that 8.8% (90 of out of 1,022 strains) is an upper-bound on the percentage of defective strains because some of these non-responsive strains may represent a special case. In total, adding the 43 strains to the 114 strains that initially failed to be constructed, 157/5804 diploid strains were not successfully produced.

### Identifying genes that impair growth

Toxic strains were identified from the YETI-NE and YETI-E collections by growth on synthetic complete media at 0 (x) and 100 nM (y) β-estradiol concentration. We first normalized our y values for unexpected changes in growth due to the addition of β-estradiol. Strains at growth extremes, the top 2% and bottom 4% at 0 and 100 nM were removed. We expect the remaining x (0 nM) and y (100 nM) values to have a 1:1 relationship. We fit a linear model: $y_{measured} = m*x_{measured}+b$. $y_{corrected} = x_{predicted}$ ($x_{predicted}=(x_{measured}-b)/m$). We can then calculate changes in growth on β-estradiol using the distance from the line equation $= abs(ax_{measured} + by_{corrected} + c) / sq\_root(a^2 + b^2)$. Genes with a distance from the diagonal larger than 2,000 were considered toxic and used in downstream analysis. This distance cutoff agreed with genes identified as toxic from YETI-E clustering analysis.

### Reversibility of overexpression phenotypes

Strains were first pinned onto SC lacking β-estradiol, and then colonies were transferred to fresh YNB plates either with or without β-estradiol. After ~ 24 h of growth, colonies were transferred again to YNB plates with or without β-estradiol to begin a 120-h time-lapse growth assay. Four conditions were tested: (i) transfer from no β-estradiol to no β-estradiol (0-0), (ii) transfer from 10 nM β-estradiol to 10 nM β-estradiol (continuous β-estradiol, 10-10), (iii) transfer from no β-estradiol to 10 nM β-estradiol (addition of β-estradiol), and (iv) transfer from 10 nM β-estradiol to no β-estradiol (removal of β-estradiol). Growth curve AUCs were calculated for each strain in each condition.

### RNA-seq

To compare the expression of Z₃pr-driven alleles to that of native promoter-driven alleles, we sequenced RNA from a subset of strains after induction with nine different doses of β-estradiol. We utilized a low-input extraction-free RNA-seq protocol, similar to that found in (preprint: Ghimire *et al*, 2021).

Strains were grown overnight in SC + ClonNAT with glutamate as a nitrogen source in a 96 deep-well plate. In the morning, cultures were diluted to an OD of 0.1 in the same media and the

culture was grown for 5 h until the cells reached log phase. Once the cells reached exponential phase, 1 μl of cells (~ 10,000 cells in total) was taken and flash-frozen in liquid nitrogen as an initial timepoint. β-estradiol was then added to the remaining cultures, and the cells were grown for an additional 30 min before the final time points were taken. All samples were frozen in 7 μl of mastermix, which was comprised of 0.8 μl of 10× lysis buffer, 3.15 μl of $dH_2O$, 1 μl of ERCC RNA spike ins, 0.05 μl of RNAse inhibitor, and 2 μl of oligo-dT primer (sequence: AAGCAGTGGTATCAACGCAG AGTACTTTTTTTTTTTTTTTTTTTTTTTTTTTTTVN). The samples were then processed as follows:

1  *Reverse Transcription*: The samples were freeze-thawed 3× and incubated at 72°C for 3 min and 4°C for 2 min to denature the RNA secondary structure and enhance oligo-dT primer binding. 12 μl of the reverse transcription mastermix (containing 4 μl 5× FS buffer, 1 μl 48 μM LNA template switch oligo (sequence: AAGCAGTGGTATCAACGCAGAGTACrGrG+G), 2 μl 10 mM dNTP mix, 2.5 μl 20 mM DTT, 0.5 μl RNAse inhibitor, and 2 μl reverse transcriptase) was added to each sample and they were incubated at 42°C for 90 min, 70°C for 10 min, and kept at 4°C until the next step.
2  *cDNA amplification*: 30 μl of a PCR mastermix (25 μl 2× SeqAmp PCR buffer, 1 μl 12 μM ISO PCR primer (sequence: AAGCAGTGGTATCAACGCAGAGT), and 1 μl seqAmp DNA polymerase, and 3 μl dH$_2$O) was added to the sample. 10 PCR cycles were then run on the samples. 95°C for 1 min, 10× of 98°C for 10 s, 65°C for 30 s, 68°C for 3 min, 72°C for 10 min, and then kept at 4°C until the next step.
3  *cDNA cleanup*: an XP bead cleanup was done at 0.9×, and 11 samples were run on the bioA to check quality and concentration. The rest of the samples were quantified using pico-green.
4  *Library prep and sample pooling*: The Nextera XT kit was used for library prep. 2 μl of each sample was pooled together, and a double XP bead cleanup was done on the pooled libraries before running on the sequencer (hiseq 4000).
5  After demultiplexing, fastq files were processed using Salmon and aligned using STAR to obtain TPMs.

## Microarrays

Experiments with phosphate-limited chemostats were performed as described in Hackett *et al* (2020). Extraction, labeling, and hybridization of RNA were performed as described in Hackett *et al* (2020). Microarrays were imaged using an Agilent SureScan G4900DA Microarray Scanner. Images were processed using Agilent Feature Extraction software and a custom R script. Fluorescence intensities were floored at a value of 50. The resulting data were hierarchically clustered (Pearson distance metric with average linkage) using Cluster 3.0 (Hoon *et al*, 2004) and visualized in R (R Core Team, 2018) using the *pheatmap* package (Kolde, 2019).

## BAR-seq

To assess competitive fitness of strains in the YETI collections, pooled cultures were propagated at low density ($OD_{600}$ < 0.05) in either SC + 2% glucose +monosodium glutamate + ClonNAT or YNB +2% glucose + monosodium glutamate + ClonNAT either with or

without addition of 100 nM β-estradiol, and > 2 × 10$^6$ cells were collected at each time point. A wild-type strain containing the Z$_3$EV module integrated at the *HAP1* locus was spiked in (3% of total cell number). For BAR-seq experiments performed in SC medium, samples were collected at $t$ = 0, 9, 18, and 36 h. In YNB medium, samples were collected at $t$ = 0, 12, 24, and 48 h. Genomic DNA was extracted using the YeaStar Genomic DNA kit (Zymo Research), and 5 ng of DNA were used to PCR amplify the barcode region with custom primers (1 min at 98°C; 20 cycles at 98°C for 30 s, 65°C for 30 s, 72°C for 45 s; 3 min at 72°C). Primers were designed as follows, where lowercase n indicates the position of the 8-bp index for multiplexing:

Forward:  5'-CAAGCAGAAGACGGCATACGAGATnnnnnnnnGT CTCGTGGGCTCGGAGATGTGTATAAGAGACAGCGGTGCGAGCGGA TC-3'.
Reverse:  5'-AATGATACGGCGACCACCGAGATCTACACnnnnnnnn nTCGTCGGCAGCGTCAGATGTGTATAAGAGACAGGCACCAGGAAC CATATA-3'.

PCR products were purified with two volumes of RNAClean XP beads (Beckman Coulter) and then quality-checked and quantified on a Bioanalyzer (Agilent). Samples were pooled in equal proportions and purified further using the DNA Clean & Concentrator kit (Zymo Research). Barcodes were sequenced with a NovaSeq 6000 Sequencer (Illumina) according to manufacturer's instructions and with addition of 30% PhiX Control v3 Library.

Samples were demuxed, and barcodes were called with BARCAS (Mun *et al*, 2016) allowing up to two barcode mismatches. To generate heatmaps, the data were floored to the background level which was determined using the barcode count distribution of the time 0 samples. The barcode abundance for each strain was then normalized to the barcode abundance of the wild-type strain and then normalized to the mean value of the three time 0 samples of the respective strain and displayed as a log2 fold change.

### Z$_3$EV and Z$_3$EB42 growth comparisons

All steps were carried out at 26°C. To test reversibility of essentiality, a panel of YETI-E diploids were pinned onto SC-URA+ClonNAT and incubated for two days. Colonies were then pinned onto sporulation medium and incubated for five days. To select for haploids containing Z$_3$EV-controlled alleles, cells were pinned onto SC-HIS-ARG-LYS-URA+canavanine+thialysine+ClonNAT medium and grown for two days in the presence of 10 nM β-estradiol. Colonies were then pinned to SC-HIS-ARG-LYS-URA+canavanine+ thialysine+ClonNAT medium containing either 0 or 10 nM β-estradiol and grown for two days. These colonies were then again pinned onto the same medium containing either 0 or 10 nM β-estradiol. Plates were imaged after two days of growth. To test reversibility of Z$_3$EB42-driven alleles, the same procedure was used, except for the addition of a higher concentration of β-estradiol (1 μM instead of 10 nM). Colonies on the perimeter of plates were removed from all analyses. The remaining colony sizes were quantified using *gitter (*Wagih & Parts, 2014*)*. We then computed the ratio of colony sizes in the presence and the absence of β-estradiol. Reversibility was defined using the following cutoffs:

Growth[No_Estradiol]/Growth[With_Estradiol] <= 0.55: reversible Growth[No_Estradiol]/Growth[With_Estradiol] > 0.55–0.75: partially reversible Growth[No_Estradiol]/Growth[With_Estradiol] > 0.75: not reversible.

Identifying $Z_3EB42$-driven alleles that had β-estradiol-dependent growth was done using the following procedure:

1   Colonies were pinned onto SC-URA+ClonNAT (as diploids) and grown at 26°C for two days.
2   Colonies were pinned onto sporulation medium and grown at 26°C for five days to select for haploids with $Z_3EB42$-driven alleles.
3   Colonies were pinned on SC-HIS-ARG-LYS-URA+canavanine+ thialysine+ClonNAT with 0, 10, 100, or 1,000 nM β-estradiol and grown for two days.
4   Images were taken and quantified using *gitter*.

### Gene Ontology

Gene Ontology (GO) enrichments were performed using GO-term Finder Version 0.86, which is made available through the *Saccharomyces cerevisiae* Genome Database (https://www.yeastgenome.org/goTermFinder).

### ROF1 Synthetic Dosage Lethality (SDL)

Strain BY6442 *MATα HAP1+::natMX::ACT1*pr-$Z_3EV$-*ENO2term ura3Δ0 can1Δ::STE2*pr-*SpHis5 his3Δ1 lyp1Δ URA3*-$Z_3$pr(*6 binding sites*)-*ROF1* was derived from the YETI-NE heterozygote, ed2531, by tetrad dissection. SGA was done at 26°C using a standard protocol (Kuzmin *et al*, 2014): mating with the deletion mutant array (DMA) and the array of strains containing temperature-sensitive alleles (TSA) on YPD (1 day), diploid selection on YPD+G418+ClonNAT (2 days), sporulation on spo + 1/4 G418 (5 days), first haploid selection on SD-his-ura-lys-arg+canavanine+thialysine (3 days), second haploid selection on SD-his-ura-lys-arg+canavanine+thialysine +G418 (2 days). For the final selection step, cells were pinned with 0.5 mm pins to SD-his-ura-lys-arg +canavanine+thialysine+G418+ ClonNAT+0 nM β-estradiol and to SD-his-ura-lys-arg +canavanine+ thialysine+G418+ClonNAT+1,000 nM β-estradiol and incubated for 2 days (0 nM β-estradiol) and 5 days (1,000 nM β-estradiol). Colony size was assessed using *gitter* (Wagih & Parts, 2014), and SDL interactions were identified by dividing normalized colony size on plates with 1,000 nM β-estradiol by normalized colony size on plates with 0 nM β-estradiol. Synthetic dosage lethal interactions were called for strains scoring < −0.08 in 2/2 replicate screens.

### Assessing expression of cells with different $Z_3$ promoters driving mNeonGreen

Yeast strains with $Z_3EV$ and $Z_3EB42$ transcriptional activators in combination with various $Z_3$ promoters were grown to early log phase in YNB medium and then diluted to $OD_{600} = 0.05$ in YNB containing a range of concentrations of β-estradiol. Cells were immediately inoculated in a 384-well imaging plate for growth at room temperature and imaged every hour with a PerkinElmer Phenix automated confocal imaging system using digital phase contrast and EGFP channels. Cells were segmented in the digital phase contrast channel, and average mNeonGreen intensity/cell was determined using the Harmony software package (PerkinElmer).

To assay the effects of β-estradiol removal, cells were grown to early log phase in YNB (containing 10 nM β-estradiol for $Z_3EV$ and 1,000 nM for $Z_3EB42$), washed 3× in PBS, and inoculated in YNB lacking β-estradiol. Imaging and analysis were done as described above. Doubling time for cells in these conditions was 4–5 h.

## Data availability

Next-generation sequencing and microarray data can be found at the Gene Expression Omnibus with accession number GSE158319: https://www.ncbi.nlm.nih.gov/geo/query/acc.cgi?acc = GSE158319

Platometer software: https://github.com/baryshnikova-lab/platometer.

**Expanded View** for this article is available online.

## Acknowledgements

The authors acknowledge Sean Hackett, Brian Feng, Chiraj Dalal, and Meng Jin for providing critical feedback on the manuscript, and Bernd Wranik for providing assistance with the *ROF1* overexpression experiment. The authors thank Andrew Bognar for discussions about *SHM2*, Jef Boeke for information and reagents associated with URS1, and Adam Baker for his design insights and contributions to figure preparation. Finally, the authors thank Jason Rogers for providing the direct amplification RNA-seq protocol, Nathaniel Thayer for advice on aneuploidy detection, and Calvin Jan for guidance with RNA-seq analysis of human datasets. This work was funded by Calico Life Sciences LLC, NIH grant RO1GM046406 (DB) with a sub-grant from Princeton University to the University of Toronto (BJA), and grants FDN-143264 (CB) and FDN-143265 (BJA) from the Canadian Institutes of Health Research.

## Author contributions

YA, HF, GT, CB, BJA, and RSM wrote the main text of the manuscript. All authors contributed to the Materials and Methods, edits, and revisions of the manuscript. YA, GK, ZL, HF, RYW, DC, MH, EHS, and RSM performed experiments. HF, GT, RYW, MU, AB, and RSM performed computational analysis. GT and AB developed the Platometer software package. YA, ZL, and DC constructed the YETI strains. CB, DB, BJA, and RSM oversaw the study.

## Conflict of interest

GK, GT, RYW, MH, ES, AB, DB, and RSM are employees of Calico Life Sciences LLC. The remaining authors declare no competing financial interests.

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
