## [Review Process File · Molecular Systems Biology]

A genome-scale yeast library with inducible expression of individual genes

Yuko Arita, Griffin Kim, Zhijian Li, Helena Friesen, Gina Turco, Rebecca Wang, Dale Climie, Matej Usaj, Manuel Hotz, Emily Stoops, Anastasia Baryshnikova, Charles Boone, David Botstein, Brenda J. Andrews, and Scott Mclsaac

DOI: 10.15252/msb.202110207

Corresponding author(s): Scott Mclsaac (rsm@calicolabs.com) , Brenda J. Andrews (brenda.andrews@utoronto.ca), Charles Boone (charlie.boone@utoronto.ca), David Botstein (botstein@calicolabs.com)

Review Timeline:

Submission Date:	6th Jan 21
Editorial Decision:	8th Feb 21
Revision Received:	12th Mar 21
Editorial Decision:	21st Apr 21
Revision Received:	27th Apr 21
Accepted:	30th Apr 21

Editor: Maria Polychronidou

Transaction Report:

Manuscript Number: MSB-2021-10207

Title: A genome-scale yeast library with inducible expression of individual genes

Thank you again for submitting your work to Molecular Systems Biology. We have now heard back from the three referees who agreed to evaluate your study. Overall, the reviewers acknowledge that the study presents a valuable resource. They raise however a series of concerns, which we would ask you to address in a major revision.

I think that the recommendations of the reviewers are rather clear. Therefore, I do not see the need to repeat the points listed below. All issues raised by the reviewers need to be satisfactorily addressed. Please contact me in case you would like to discuss in further detail any of the issues raised.

Reviewer #1:

In this paper, Arita et al expand on earlier work engineering an estradiol-responsive transcription factor (ZEV) and associated promoter (ZEVpr) in yeast. Here they report a strain collection (YETI) that encompasses most yeast genes (including essential and non-essential genes) under control of the inducible promoter system. They characterize this new collection to determine how

overexpression of each gene affects growth, fitness and relate these effects to gene expression. The authors present a series of studies that demonstrate applications of this new resource. In addition, the authors develop a variant of the inducible system that has reduced basal expression.

Overall, this is a well-written paper that reports a novel and valuable resource for yeast systems biology. The design and construction of this collection is sound, and the experimental analysis is well performed and provides sufficient information for other groups to use these strains. The authors also provide some general insights into genome-wide perturbation screens with relevance to other systems including human studies making this study of broad interest. We provide suggestions for additional minor experimental and analytical work that should be performed and clarifications to the text and figures that should be incorporated prior to publication.

It would be helpful to have an experimentally determined rate at which strains that exhibit dosage toxicity acquire suppressor mutations. This rate is an important consideration for pooled fitness assays. It would also be interesting to know whether suppressor mutations for dosage toxicity result from loss of the ATF, promoter mutations, or are unlinked to the heterologous system.

The reversibility of induction is not clearly defined. Presumably, rapid reversibility results from rapid degradation of the induced gene product following removal of the inducer where non-reversibility would result from persistence of the gene product. Is there a relationship between reversibility and known protein half lives? Fig S11 would be augmented by placing TIP41-GFP and SGS1-GFP under Z3pr control to see what exactly is going on in terms of protein abundance after shifting to non-inducible media.

Can the authors resolve whether the 67% of essential genes that continue to grow in the absence of estradiol once they are induced is a result of continued expression of the gene or inheritance of the gene product?

The authors note that <5% of the strains they have constructed are "defective". It's not clear if this includes strains that failed construction. The authors should explain why these were not successful, as many of the corresponding strains appear in the yeast deletion collection. Is it an issue of transformation efficiency for some genes (perhaps due to the difference in strain background) or is there some biological basis that prevents these genes from functioning under inducible control.

The rationale for the E. coli analysis presented in Fig3B is not clear. No information about this analysis is present in the methods section and it isn't clear if it's performed on a per-gene or per-transcript basis. Per-gene makes more sense given the focus on essential genes, but per-transcript makes more sense based on the experimental design.

The data analyses regarding studies in human cells (i.e. Fig S14) should be described in the results section. No methods are provided for the analysis presented in Fig S14.

There are a number of applications that use either the haploid or diploid versions of the strain collection - it would be very helpful to have haploids or diploids more clearly labeled in the figures.

It would be helpful to have a supplemental figure with examples of the growth curve measured for AUGC. What are the AUGC units and why is this metric preferable over growth rate or final yield. Given the importance of this metric, a clearer explanation is critical.

The methods section lacks important details, for example, many details are omitted in favor of

colloquial names like 'Z3EV diploid maintenance plates'. Some details in the methods section are inconsistent with other text (i.e. fig1 in the methods claims 'overnight at 30C in minimal medium [...] then diluted into fresh medium and grown at 30C to a density of 6×10^6 cells/mL' and the corresponding figure legend claims 'YNB for 18 hours'. Methods are noted for '(Fig 1D)' which does not exist. In general, the methods section could use a careful check and some rewriting.

There are researchers that use different amounts of GAL for induction studies, so it is not clear that the GAL1 promoter system is a "non-graded induction".

The differences in transcript levels (TPM) of the 19 induced non-essential genes is striking (2 orders of magnitude). Codon usage affects transcript stability and it would be interesting to test if this is correlated with variation in expression levels.

It is surprising that 33% of essential genes grow in the absence of inducer and also demonstrate dosage-independent growth. Is there evidence that these genes are actually overexpressed (or evidence that they have not acquired a suppressor/escape mutation).

The Z3 promoter is placed upstream of the ORF, but the native promoter is not deleted. Are the native promoters still active? For example, can GAL transcripts that have the Z3 promoter still be activated by Galactose? Would this residual activity explain the observed basal activity in the absence of inducer?

For RNA-seq analysis: "1uL of cells" for t0? That does not sound right

In figure 1A, placing HSP90 on top of hER would prevent anyone from misunderstanding that HSP90 is translocating to the nucleus.

In figures 1B and 1C it would be more interesting to plot the single cell measurements. Is there any evidence for bimodality in the response, which was reported in an earlier paper using this system. Given that flow cytometry was performed, the single cell measurements are available.

There is a school of thought that holds that one should not plot continuous data in bins. Can figure 3A not be plotted using continuous x-axis values?

Given the experimental design for the barseq study it would be more appropriate to perform a statistical analysis that tests for differences in response using untransformed count data and methods for significance testing such as those implemented in DEseq or edgeR.

It would be easier to interpret figure 7 if the samples were on the same plot.

In figure S2, it would be easier to see if the induction kinetics are the same by including the different genotypes in the same plot.

There is no explanation of how aneuploidy was determined from sequence data. Presumably it is read depth.

Why is there no variant analysis to test whether these strains have acquired mutations (point mutations or CNVs) during construction? These data are underutilized in the paper as they would also provide confirmation of the integrity of the Z3pr and Z3EV gene.

Reviewer #2:

Arita et al. report a yeast strain library where each strain is engineered to have an estradiol-inducible gene in its native genomic loci. The team established 5687 strains based on the Z3EV estradiol-inducible gene expression system. They found that some essential gene strains did not display expected growth defects under no estradiol condition. This was presumably because their background expression leak levels are similar to those of which expression levels are low in the wild type strain. Thus, they also established another library for essential genes with a dialed down version of the estradiol-inducible gene expression system (Z3EB42). Overall, they performed careful analyses to characterize the libraries. However, I would suggest more data analyses to provide full characteristics of the collection to the community as described below. The specificity of the pooled Bar-seq analysis seemed much lower than that of the plate-based assay. I was not impressed by this demonstration. More careful quantitative comparisons with the previously established GAL4 promoter-based libraries need to be demonstrated.

Major comments:

It's great that the authors show the off kinetics of the Z3EV system upon the removal of estradiol in Fig 1c, but in Fig S1, the induction of fluorescent reporter is not saturated. It is important to provide full characteristics of the system, especially on how long it takes so the induction reaches the saturation level and its amplitude. It's also nice that the authors show gene induction levels of different genes with different inducer concentrations, but how the on and off kinetics of each gene behave by applying and removing the inducer? Do they all look similar to that of the eGFP experiment?

Fig S1: the fluorescence induction level is not saturated at 9 hrs.

Was Fig S2 performed by RNA-seq or by RT-qPCR? Can the author comment on the effects of these gene inductions on the transcriptome landscape if done by RNA-seq?

Figure 2: What is the model to explain Cluster 4? Is it likely that beta-estradiol is toxic when a certain gene is overexpressed? Induction level analysis of few target genes in every cluster across different estradiol concentrations seems necessary to discuss the potential mode of actions behind each cluster. It was not clear if Fig S3 (18 genes) covered this.

Some essential gene strains did not show phenotypic defects without beta-estradiol. The authors suggest this could be because their background leak levels were similar to wild type, and those genes tended to be lowly expressed in wild type. These are great observations and discussion but triggers another question how many YETI-NE strains are incompetent because of the same reason. Careful investigation on this is necessary to provide the resource to the community.

I don't understand why Fig 3b (E coli result) is more prioritized than Fig S4.

Page 10: "We found that genes whose proteins have a high percentage of IDRs were more toxic for non-essential, but not essential, genes" What is the sample size? Is this statistically significant? All the panels in Fig S7 need statistical tests if any difference is claimed.

Fig S8a and b are redundant and very complicated. For example, the Venn diagram represents the intersection of barFLEX and YETI has $62 + 111 = 173$ but the panel b says "all shared" are 62 and barFLEX and YETI are 111. Did you exclude the "all shared" from barFLEX and YETI in the panel b?

What is the point of showing both of the panels? Furthermore, rather than redefining the quantitative growth values into the discrete labels of "impaired growth" or not, the authors should show scatter plots of growth values from different assays with statistical tests. Why was this avoided?

Fig 5 is just a heatmap conversion of the raw data and doesn't provide much information. Also, many discussions here are based on data presented by large supplementary tables and hard to understand. Figure displays would greatly help the understanding of what the authors are claiming. For example:

"The resulting data were normalized to data collected at time 0, then hierarchically clustered to look for general trends (Table S7A)" This doesn't represent any hierarchical clusters.

"Indeed, we saw strong GO enrichment for biosynthetic processes, including amino acid and organic acid biosynthesis, and the average Aux score for genes in this cluster was relatively high (Table S7B)" This could be a small panel in Fig 5 with statistical tests.

Fig S9: It looks like only a small fraction of the extremely enriched and depleted strains in the Bar-seq assay agreed with the on-plate data, and this assay could produce many false positives. If the Bar-seq assays of different estradiol concentrations and those of the on plate assays are compared, respectively, how do they look like?

Page 12: "To explore this approach with the YETI collection, we chose ROF1, which encodes a putative transcription factor containing a WOPR DNA-binding domain and whose induction with β -estradiol inhibits growth, both on plates and in BAR-seq experiments [73,74] (Figure 5, Figure 6, Table S6A, Table S7A)." Where is ROF1 in Fig 5? Why Fig 6 is cited here?

Page 13: "ROF1 also showed SDL interactions with numerous genes involved in chromatin remodeling and general transcription, including multiple members of the Swr1 complex, Ino80 complex, Rpd3L complex, NuA4 complex, and COMPASS, which presumably reflects its role as a transcriptional repressor and a regulator of other TFs." How do their transcriptomes look like? Do they agree with what the authors observed in Fig 6?

Page 13: "These data suggest that many essential genes under the control of Z3E V may continue to be expressed in the absence of β -estradiol for numerous cell doublings." Are they also lowly expressed genes?

Re: "reversibility," is there any epigenetic memory effect observed? Like, do some epigenetic memories get triggered by overexpression of a target gene, which do not allow the cells to revert back even after removing estradiol or affect the secondary induction? What I heard about this collection a while ago, I thought one strong application would be a genome-wide study of such epigenetic memory effects.

Minor comments:

Page 4: "To select for strains that carried a wild-type HAP1 gene, the gene encoding the Z3E V transcription factor was integrated next to the repaired HAP1 together with a natMX selectable marker in Y14789." What do you mean by "repaired"? Doesn't Y14789 encode a functional HAP1?

Y14789 has a different genetic background from the BY strain, the background of the deletion collections. Is this problematic for genetic interaction assays using the SGA methodology?

I thought it would be nicer if authors encode a transcribing DNA barcode (RNA barcode) under the Z3 promoter such that the gene induction levels can be massively quantified by deep sequencing of RNA barcodes and their genomic DNA.

Having a discussion on their system in contrast to CRISPRa would be nice. Some people would quickly think constructing a gRNA library for CRISPRa is way easier than creating these strain libraries.

Page 13: "However, 16 of the 24 (67%) Z3E V strains grew in the absence of β -estradiol, even though their initial growth had been β -estradiol-dependent." Fig S10 needs to be cited here.

Reviewer #3:

In this paper, the authors generate a genome-wide collection of yeast strains. Each of these strains contains an inducible promoter swapped substituted for a different gene's native promoter. These inducible promoters are ZEV variants that are inducible with estradiol. This collection is a powerful and versatile resource because yeast is not naturally responsive to estradiol, meaning genes can be induced in different environments and without triggering a general response to the estradiol. This whole system is much more precise and titratable than alternatives, such as galactose induction.

This paper is a nice achievement, as generating and checking all of these strains was undoubtedly a substantial amount of work. This collection seemed to have been produced in a careful and rigorous manner. The experiments that are described show how interesting biological insights can be obtained using the YETI collection. I should also say, this paper was written with simple and easy to read language. I felt anyone could read and mostly understand this manuscript, which will help ensure this paper has a broad audience.

I do not have any substantive comments on this paper. The following suggestions are more nitpicky and are simply places where I noticed a bit more information or clarity might be helpful.

1. p4, RCY1792. Is this an S288c derivative? The name was unfamiliar and it would be good to connect this to the commonly used strains if possible.
2. p5, 'TPS2 or LEU2' should probably be flipped to 'LEU2 or TPS2' given what comes later in the sentence.
3. p5, The integration of the Z3 promoters is described only superficially in the main text. Details are in the methods. However, a bit more information is probably necessary in the main text.
4. p8, The E. coli discussion was interesting, but is there no comparable data in yeast? For example, the Steinmetz lab has done genome-wide CRISPRi in yeast.
5. p9, 'strong selection pressure resulting in aneuploidy'. The aneuploidies happen and are then selected. It is not that selection generates aneuploidies, unless there is something I am missing.

6. p14, The main text might benefit from a brief description of why reversibility might occur.

7. f5, The panel at the top is a little confusing in my opinion. Is there a more intuitive way to draw the sampling scheme for Barcode sequencing?

Reviewer #1:

In this paper, Arita et al. expand on earlier work engineering an estradiol-responsive transcription factor (ZEV) and associated promoter (ZEVpr) in yeast. Here they report a strain collection (YETI) that encompasses most yeast genes (including essential and non-essential genes) under control of the inducible promoter system. They characterize this new collection to determine how overexpression of each gene affects growth, fitness and relate these effects to gene expression. The authors present a series of studies that demonstrate applications of this new resource. In addition, the authors develop a variant of the inducible system that has reduced basal expression.

Overall, this is a well-written paper that reports a novel and valuable resource for yeast systems biology. The design and construction of this collection is sound, and the experimental analysis is well performed and provides sufficient information for other groups to use these strains. The authors also provide some general insights into genome-wide perturbation screens with relevance to other systems including human studies making this study of broad interest. We provide suggestions for additional minor experimental and analytical work that should be performed and clarifications to the text and figures that should be incorporated prior to publication.

1-1. It would be helpful to have an experimentally determined rate at which strains that exhibit dosage toxicity acquire suppressor mutations. This rate is an important consideration for pooled fitness assays. It would also be interesting to know whether suppressor mutations for dosage toxicity result from loss of the ATF, promoter mutations, or are unlinked to the heterologous system.

We expect that our YETI strains will acquire mutations (suppressor or otherwise) at the same rate as other strains. Indeed, there is general agreement of overexpression phenotypes between our data and published experiments (Figure EV3 and Dataset EV6), suggesting that the YETI strains have not acquired suppressors. Because researchers can control gene expression levels (and therefore toxicity) with estradiol concentration, we predict that suppressor accumulation will not be a problem. Furthermore, we have constructed the collection as heterozygous diploids in which strains contain two copies of a particular target gene (one that is inducible and one in that is under the control of the native promoter), which means the strains are unlikely to accumulate suppressors.

1-2. The reversibility of induction is not clearly defined. Presumably, rapid reversibility results from rapid degradation of the induced gene product following removal of the inducer where non-reversibility would result from persistence of the gene product. Is there a relationship between reversibility and known protein half lives? Fig S11 would be augmented by placing TIP41-GFP and SGS1-GFP under Z_3pr control to see what exactly is going on in terms of protein abundance after shifting to non-inducible media.

This is an interesting question, and we have explored it; however, we do not have a definitive answer at this time. We see no inverse correlation between reversibility of expression phenotypes and RNA stability (using 3 different datasets [Wang 2002, Geisberg 2014, and Chan 2018]) or protein stability (using 3 different datasets [Belle 2006, Christiano 2014, and Martin-Perez 2017]), or transcript abundance. However, the published datasets also don't show quantitative agreement with one another. For each of the half-life datasets, we extracted the genes for which we have reversibility data. For our genes of interest, the best Pearson correlation between RNA stability datasets was ~0.43 (R^2 of a linear fit was 0.19), and the best Pearson correlation protein stability datasets was ~0.33 (R^2 of a linear fit was 0.11). We are left to speculate that reversibility is related to multiple biological parameters, which will require future work to investigate fully.

1-3. Can the authors resolve whether the 67% of essential genes that continue to grow in the absence of estradiol once they are induced is a result of continued expression of the gene or inheritance of the gene product?

This is also an interesting experimental idea, and is not something we explicitly tested. We've shown that native expression is a critical variable, and that by weakening the Artificial Transcription Factor as well as its target promoter that we can make a more reversible system. But there is a cost. Increased reversibility requires decreased transcriptional activation in the presence of saturating levels of inducer. This result argues that - in the case of Z₃EV - there is continued transcriptional expression of the gene after the inducer is removed.

1-4. The authors note that <5% of the strains they have constructed are "defective". It's not clear if this includes strains that failed construction. The authors should explain why these were not successful, as many of the corresponding strains appear in the yeast deletion collection. Is it an issue of transformation efficiency for some genes (perhaps due to the difference in strain background) or is there some biological basis that prevents these genes from functioning under inducible control.

We have elaborated on this in the **Methods** section as follows:

Strain construction statistics

Correct insertion of the URA3::Z₃pr cassette was confirmed by PCR for all strains. Only 114/5804 strains (2%) failed construction, even after multiple attempts, as assessed by absence of PCR product in the confirmation assay. Some of the failed ORFs are close to transposable elements or other repetitive sequences, which could result in the absence of a specific band in our confirmation PCR (Dataset EV2). We analyzed growth on β -estradiol to further assess whether strains confirmed by PCR had acquired defects (e.g. through PCR or recombination) such that the Z₃pr was unable to induce expression of the gene. Following our initial construction, we identified 90 strains that had little or no growth at all concentrations of β -estradiol. We attempted to re-make 54 of these strains and were able to construct 47 that had growth at some concentration of β -estradiol (Dataset EV3, reconstructed strains), suggesting that most of the 90 that initially had little or no growth in response to β -estradiol were defective for their original

construction. We believe that 8.8% (90 of out of 1,022 strains) is an upper-bound on the percentage of defective strains because some of these non-responsive strains may represent a special case. In total, adding the 43 strains to the 114 strains that initially failed to be constructed, 157/5804 diploid strains were not successfully produced.

1-5. The rationale for the *E. coli* analysis presented in Fig3B is not clear. No information about this analysis is present in the methods section and it isn't clear if it's performed on a per-gene or per-transcript basis. Per-gene makes more sense given the focus on essential genes, but per-transcript makes more sense based on the experimental design.

The *E. coli* analysis involved combining two published datasets: 1) a CRISPRi screen, and 2) a gene expression dataset of *E. coli* (GSE67218). In the figure legend, we wrote, “*E. coli* strains in which CRISPRi targets essential genes are more likely to grow if the targeted essential gene is lowly expressed. Boxplots show the distribution of gene expression levels for *E. coli* genes tested in a CRISPRi screening experiment [61]. The genes are grouped into essential genes whose repression by CRISPRi inhibits growth (does not grow) or fails to inhibit growth (grows). Gene expression data from Gene Expression Omnibus (GSE67218). RPKM (Reads Per Kilobase of transcript, per Million reads) median values are 166 and 503 for the “Grows” and “Does not grow” classes, respectively.” We thought it would be redundant to also write this in the Methods. We’ve added the text in red to note the definition of RPKM. The RPKM values were reported in GSE67218.

Why did we add this analysis? In the Results section, we wrote, “*To test if native expression level is also an important factor for achieving conditional growth for essential genes in organisms other than yeast, we investigated a recent CRISPRi pooled screen from Escherichia coli (Wang et al, 2018). A useful feature of E. coli for this analysis is the availability of a deletion mutant collection from which a core set of essential genes has been determined (Wang et al, 2018; Baba et al, 2006).*”

1-6. The data analyses regarding studies in human cells (i.e. Fig S14) should be described in the results section. No methods are provided for the analysis presented in Fig S14.

All of the info for reproducing the plots is presented in the figure legend. The RNA-seq data comes from Encode (<https://www.encodeproject.org/experiments/ENCSR545DKY/> [GSE88351 on GEO]). The list of essential genes is from Hart et al. (2017, G3). We've decided to keep these data in the Discussion section. The analysis in the Results section focuses on cases where we have gold standard essential-gene lists. The analysis of human data is more speculative due to the lack of such gold standards. We think it is worth discussing and hope that its inclusion in the current manuscript will spur future work.

1-7. There are a number of applications that use either the haploid or diploid versions of the strain collection - it would be very helpful to have haploids or diploids more clearly labeled in the figures.

We thank the reviewer for pointing this out as something that needs clarification for readers. All assays were performed using haploids. Even for the experiments in **Figure 2**, in which we started with diploids, the growth measurements were done in haploid strains. In the legends of Figures 2, 4, and 5 we now specifically refer to haploids.

1-8. It would be helpful to have a supplemental figure with examples of the growth curve measured for AUGC. What are the AUGC units and why is this metric preferable over growth rate or final yield. Given the importance of this metric, a clearer explanation is critical.

We have added a new figure (Figure EV2, included below) to show growth curves of the YETI for a handful of YETI-E strains highlighted in Figure 2 (*ALG1*, *ULP2*, *CDC48*, and *ADE13*). AUGCs are in units of pixels². We use AUGC because it captures total growth and is less sensitive to parametric fitting than other metrics. We note in the Methods, “For each colony, the area under the growth curve (AUGC), which integrates the main three aspects of yeast population dynamics (duration of lag phase, exponential growth rate and carrying capacity), was used as a global estimate of growth efficiency.” In the main text, we’ve added the following text: *“Colony growth was measured over time and growth curves were quantified by determining the Area Under Growth Curve (AUGC). We utilized this metric because it is insensitive to specific data parametrizations.”*

1-9. The methods section lacks important details, for example, many details are omitted in favor of colloquial names like 'Z3EV diploid maintenance plates'. Some details in the methods section are inconsistent with other text (i.e. fig1 in the methods claims 'overnight at 30°C in minimal medium [...] then diluted into fresh medium and grown at 30°C to a density of 6 x 10⁶ cells/mL' and the corresponding figure legend claims 'YNB for 18 hours'. Methods are noted for '(Fig 1D)' which does not exist. In general, the methods section could use a careful check and some rewriting.

While we gave “colloquial” names for different types of media, we did so carefully and deliberately. The names we give different media types in the Main Text/Methods are detailed in the Appendix under “Media Recipes” along with protocols for making them. For 'Z3EV diploid maintenance plates', as an example:

Z3-E and Z3-NE Diploid Maintenance Medium

YPD+ClonNAT

For 1L

- In 900 mL water, add
 - 10 g Yeast Extract
 - 20 g Bacto Peptone
 - 20 g Agar
- autoclave
- Add 100 mL 20% glucose (w/v)
- Mix well and cool to ~60°C
- Add 1 mL 1000X ClonNAT (100mg/mL; final concentration 100 µg/mL)

For clarification, we’ve added “The yeast strains and plasmids used in this study are listed in the **Appendix**. All media recipes are provided in the **Appendix**.” to the beginning of the Methods section.

Additionally, we’ve gone through the Methods section carefully for the revision. For the FACS experiments, we made a mistake in the legend of our original submission. Overnight cultures were diluted and then grown to 6 x 10⁶ cells/mL (so that the cells were actively dividing). Cells were then incubated with estradiol for 6 hr. We appreciate the reviewer noticing our mistake.

1-10. There are researchers that use different amounts of GAL for induction studies, so it is not clear that the GAL1 promoter system is a "non-graded induction".

We have removed the reference to “non-graded induction”.

1-11. The differences in transcript levels (TPM) of the 19 induced non-essential genes is striking (2 orders of magnitude). Codon usage effects transcript stability and it would be interesting to test if this is correlated with variation in expression levels.

We agree with the reviewer that this is an interesting idea. It is worth noting that, elsewhere, codon usage has been shown to be relevant for gene expression levels in *Saccharomyces*

cerevisiae (e.g. Yang et al., 2021; PMID: 33410890, Zur and Tuller 2013; PMID: 24564391). Since alleles in the YETI library use the same codons as wild-type yeast, we believe the published results could be integrated with our data in future work.

1-12. It is surprising that 33% of essential genes grow in the absence of inducer and also demonstrate dosage-independent growth. Is there evidence that these genes are actually overexpressed (or evidence that they have not acquired a suppressor/escape mutation).

The reviewer is referencing the Cluster 1 genes. Many of the Cluster 1 genes show a mild toxicity upon overexpression (Figure 2, quantified in Dataset EV3). We now mention this in the text (see below) since these results provide evidence that the genes are indeed overexpressed.

“Most remaining strains in the YETI-E collection had a dosage-independent growth response, including a large set of strains (33%) that grew well in the absence of β -estradiol (Cluster 1); many of these had a mild growth impairment in high β -estradiol concentrations. A smaller set (9.2%) grew in the absence of inducer, with more substantial growth inhibition at higher concentrations (Cluster 2 – dosage toxicity).”

1-13. The Z3 promoter is placed upstream of the ORF, but the native promoter is not deleted. Are the native promoters still active? For example, can GAL transcripts that have the Z3 promoter still be activated by Galactose? Would this residual activity explain the observed basal activity in the absence of inducer?

This is an interesting point. It would be hard to prove that the sequence of the native promoter of the native gene has absolutely no effect on the Z₃pr-driven ORF. The proposed GAL experiment is an interesting idea, but it would be confounded because Z₃pr would become de-repressed on media that only contained galactose as a carbon source. From an engineering perspective, we've utilized the Mig1 binding site in Z₃pr to maximize dynamic range on glucose-containing media. Our promoter system is still inducible with estradiol in strains grown on galactose, but the basal expression level is higher.

There's ~2000 base pairs between the native ORF and the Z₃pr-driven ORF and within those 2000 base pairs is also the URA3 gene. From work in Fred Winston's lab and global ChIP-ChIP experiments (i.e., Harbison et al.), TF binding sites cluster near the ORF (<300-500bp from the first ATG). When binding sites are moved ~700 base pairs upstream of the ORF they lose their potency -- see Figure 1 in Dobi et al. (Molecular and Cellular Biology, 2007). For these reasons, we believe that the native promoters have minimal regulatory effect on Z₃pr-regulated ORFs. Our choice to leave them in the genome was to minimize disruption to the regulation of neighboring genes.

1-14. For RNA-seq analysis: "1 \$\mu\$ L of cells" for t0? That does not sound right

This was not a typo. We used ~10,000 cells for RNA amplification. We've added some text as well as a reference to clarify: *“To compare the expression of Z₃pr-driven alleles to that of native*

promoter-driven alleles, we sequenced RNA from a subset of strains after induction with nine different doses of β -estradiol. We utilized a low-input extraction-free RNA-seq protocol, similar to that found in (Ghimire et al)...Once the cells reached exponential phase, 1 μ L of cells (~10,000 cells in total) were taken and flash frozen in liquid nitrogen as an initial timepoint.”

1-15. In figure 1A, placing HSP90 on top of hER would prevent anyone from misunderstanding that HSP90 is translocating to the nucleus.

We thank the reviewer for this suggestion that will help clarify how the system works. We've changed the figure as shown below.

1-16. In figures 1B and 1C it would be more interesting to plot the single cell measurements. Is there any evidence for bimodality in the response, which was reported in an earlier paper using this system. Given that flow cytometry was performed, the single cell measurements are available.

We have previously published single-cell data on the ZEV system (different strain background and slightly different constructs): see Figure 6 in McIsaac et al. (JoVE, 2013), <https://www.jove.com/t/51153/rapid-synthesis-screening-chemically-activated-transcription-factors>

In that study, there was no evidence of bi-modality. Interestingly, however, with these new constructs in a new strain background, the ZEV system appears more sensitive (i.e, it responds to lower concentrations of estradiol). This could be due to the inclusion of an Eno2 terminator downstream of Z3EV and/or its new location in the genome.

However, we aren't familiar with any published evidence of bimodality with the ZEV system. We show below a handful of the single-cell measurements that demonstrate the graded nature of induction. There is no evidence of bi-modality. At intermediate levels of induction, the variance of the distribution does increase slightly. Following the reviewer's suggestion, we've added a new supplemental figure (**Appendix Figure S1A**) to highlight the single-cell data.

1-17. There is a school of thought that holds that one should not plot continuous data in bins. Can figure 3A not be plotted using continuous x-axis values?

We only binned the data to make our point more clearly. The significance of the result doesn't depend on binning. The Spearman correlation is -0.495 with a p-value < 2.2e-16. We have added the following text to the manuscript: *“With unbinned data, the Spearman correlation between transcript level and AUGC (at 0 nM β -estradiol) was -0.49 (p-value < 2.2 x 10⁻¹⁶).”*

1-18. Given the experimental design for the barseq study it would be more appropriate to perform a statistical analysis that tests for differences in response using untransformed count data and methods for significance testing such as those implemented in DEseq or edgeR.

Clustering by log2-transformed data highlights biologically interpretable clusters with different dynamic patterns. We expect that other analyses can highlight new features of biological importance in the data, and we believe other researchers can analyze these data in other ways to reveal additional insights beyond what we've presented.

1-19. It would be easier to interpret figure 7 if the samples were on the same plot.

We agree with the reviewer that Figure 7 could be clearer. The expression outputs for these systems span ~2 orders of magnitude. To improve clarity, we've increased the boldness of the line that shows how the max y-axis values on the right plots correspond to the y-axis values on the left plots.

1-20. In figure S2, it would be easier to see if the induction kinetics are the same by including the different genotypes in the same plot.

These are dose-response curves. We've included them on separate plots because the TPM values range widely from gene to gene. We are trying to point out that there is a qualitative similarity in the dose responses of Z₃pr-controlled genes even when the total number of TPMs can vary from gene to gene. Given the wide range of TPM values, we believe that this point is most clearly by having a separate plot for each dose-response curve.

1-21. There is no explanation of how aneuploidy was determined from sequence data. Presumably it is read depth.

We thank the reviewer for pointing out this oversight. Indeed, aneuploidy was determined by simply using read depth. Genomic DNA was extracted and prepared for whole genome sequencing using the Nextera XT DNA library. Read depth over all chromosomes was plotted using Python to detect instances of aneuploidy visually. We've added the following text to the Methods: *“Cells were grown overnight in a diluted culture until log phase was reached in the morning. Samples were first processed using the YeaStar Genomic DNA kit (Zymo #D2002) to obtain genomic DNA and prepared for whole genome sequencing using the Nextera DNA Flex Library prep kit (Illumina #20018705). Samples were sequenced on the NovaSeq 6000. Read depth over all chromosomes was then plotted using Python to detect instances of aneuploidy.”*

1-22. Why is there no variant analysis to test whether these strains have acquired mutations (point mutations or CNVs) during construction? These data are underutilized in the paper as they would also provide confirmation of the integrity of the Z3pr and Z3EV gene.

The above analysis did reveal CNVs in SGA-pathway strains. We note this in Dataset EV1. Specifically, we found a partial duplication of chromosome 12 in the *MET17* YETI strain. This CNV contained the *MET17* open reading frame (ORF). We performed sequence confirmation for a subset of barcodes, as well as the junctions between synthetic promoters and target ORFs. While sequencing the collection and performing variant analysis could provide some information, extensive variant analysis was beyond the scope of this initial study. The expectation is that, for the majority of strains, standard lithium acetate transformation and basic yeast husbandry will not result in substantial genomic alterations.

Reviewer #2:

Arita et al. report a yeast strain library where each strain is engineered to have an estradiol-inducible gene in its native genomic loci. The team established 5687 strains based on the Z₃EV estradiol-inducible gene expression system. They found that some essential gene strains did not display expected growth defects under no estradiol condition. This was presumably because their background expression leak levels are similar to those of which expression levels are low in the wild type strain. Thus, they also established another library for essential genes with a dialed down version of the estradiol-inducible gene expression system (Z₃EB42). Overall, they performed careful analyses to characterize the libraries. However, I would suggest more data analyses to provide full characteristics of the collection to the community as described below. The specificity of the pooled Bar-seq analysis seemed much lower than that of the plate-based assay. I was not impressed by this demonstration. More careful quantitative comparisons with the previously established GAL4 promoter-based libraries need to be demonstrated.

Major comments:

2-1. It's great that the authors show the off kinetics of the Z3EV system upon the removal of estradiol in Fig 1c, but in Fig S1, the induction of fluorescent reporter is not saturated. It is important to provide full characteristics of the system, especially on how long it takes so the induction reaches the saturation level and its amplitude. It's also nice that the authors show gene induction levels of different genes with different inducer concentrations, but how the on and off kinetics of each gene behave by applying and removing the inducer? Do they all look similar to that of the eGFP experiment?

The kinetics of induction (at the transcript level) are saturating at ~10 min following addition of 1 μ M estradiol with the ZEV system. We've pasted Appendix Figure S1 from Hackett et al. (Molecular Systems Biology, 2020) below that shows the transcriptional response of 100+ TFs following estradiol induction.

ROF1 induction shows similar kinetics following estradiol induction in the Z3pr-*ROF1* strain present in the YETI collection (see below). Data from two independent replicates are shown (the data for generating this plot are in Dataset EV8).

2-2. Fig S1: the fluorescence induction level is not saturated at 9 hrs.

Based on the analysis shown in response to 2-1, we've decided to remove this figure in the re-submission.

2-3. Was Fig S2 performed by RNA-seq or by RT-qPCR? Can the author comment on the effects of these gene inductions on the transcriptome landscape if done by RNA-seq?

The analysis was done by RNA-seq. Based on the reviewer's comment we looked at the transcriptome landscapes. Recall that strains were induced with different amounts of inducer for only thirty minutes. The only result of this analysis is that Bat1 represses a variety of amino acid biosynthetic genes (see clusters below). In the absence of inducer, these genes were up-regulated (red) compared to all strains in the panel (the TPM value for each gene was median normalized). The genes below are the only ones that show clear dose-dependent repression by Bat1. The dose of the inducer is increasing from left to right. Every *HIS* gene except *HIS6* as well as all of the *ILV* genes (involved in biosynthesis of isoleucine from threonine) are repressed by Bat1 induction. We'd like to keep the discussion of these results in the Main Text to a minimum, since these results are not the focus of our paper. To that end, we've added a new **Appendix Figure S2** and the following text: *Additionally, since we used RNA-seq, we explored the transcriptome landscape of these strains. Bat1 induction resulted in the repression a variety of amino acid biosynthesis genes in a dose-dependent fashion, including all of the ILV (IsoLeucine-plus-Valine requiring) genes, which are upstream of Bat1 and are members of the superpathway of branched chain amino acid biosynthesis (Appendix Figure S2). Since Bat1 catalyzes the terminal reactions in this superpathway, the transcriptional responses are consistent with end-product inhibition.*

2-4. Figure 2: What is the model to explain Cluster 4? Is it likely that beta-estradiol is toxic when a certain gene is overexpressed? Induction level analysis of few target genes in every cluster across different estradiol concentrations seems necessary to discuss the potential mode of actions behind each cluster. It was not clear if Fig S3 (18 genes) covered this.

Our data suggest a simpler model where these genes themselves are toxic when overexpressed. In support of this, 26/46 of the genes in Cluster 4 have been identified as toxic in one or more of three overexpression toxicity datasets (Gelperin, Sopko, Douglas), none of which used estradiol. We've added the following text in red: *“A second smaller cluster showed a similar initial behavior, with growth depending on the presence and concentration of inducer, but with growth inhibition at higher estradiol concentrations, indicating dosage toxicity (Cluster 4 with 4.7% of strains). Twenty-six out of 46 genes in Cluster 4 have been shown to be toxic upon overexpression in one or more plasmid collections under control of the GAL1 promoter [10,11,57]. In total, more than half (53.7%) of the YETI-E strains exhibited β -estradiol-dependent growth that is ‘tunable’ by inducer concentration.”*

2-5. Some essential gene strains did not show phenotypic defects without beta-estradiol. The authors suggest this could be because their background leak levels were similar to wild type, and those genes tended to be lowly expressed in wild type. These are great observations and discussion but triggers another question how many YETI-NE strains are incompetent because of the same reason. Careful investigation on this is necessary to provide the resource to the community.

In response to this comment, we'd like to discuss two applications of YETI strains: 1) conditional growth of cells with essential genes, and 2) turning on individual alleles (like *ROF1*) to study regulatory networks. For the latter experiments, YETI provides the *ideal* system. The Z₃pr-regulated allele has reduced expression as compared to the native allele, and following induction, molecular changes can be tracked dynamically. For 19 out of 19 genes we tested, the Z₃pr-driven allele had lower expression than the native allele in the absence of estradiol. In a previous report (Hackett et al., Molecular Systems Biology, 2020) it was noted that roughly 90% of Z₃pr-driven TFs had lower expression than the native alleles. This is noteworthy because TFs as a class are, on average, lowly expressed.

For conditional growth analysis, our focus is on the YETI-E strains. YETI-NE strains are not expected to have severe growth defects in the absence of an inducer. From the perspective of phenotypes, it is likely that many YETI-NE alleles will either have a complete or a less severe form of a true-deletion phenotype in the absence of estradiol. Specifically, we found detectable auxotrophy for 38 out of 42 genes defined by SGD to be auxotrophic (Figure 4). Thus, YETI-NE genes were lowly expressed enough when placed under Z₃pr control to show a growth phenotype in the absence of estradiol.

2-6. I don't understand why Fig 3b (E coli result) is more prioritized than Fig S4.

We have moved the data from Figure S4 to the main text.

2-7. Page 10: "We found that genes whose proteins have a high percentage of IDRs were more toxic for non-essential, but not essential, genes" What is the sample size? Is this statistically significant? All the panels in Fig S7 need statistical tests if any difference is claimed.

We used the nonparametric Kolmogorov-Smirnov test if these distributions were different. For the non-essential genes, the distributions were significantly different (p -value $< 2e-16$). For essential genes, the distributions were not significantly different. We've now included the p -values and test information in the figure. We've added the sample sizes to the figure legend.

2-8. Fig S8a and b are redundant and very complicated. For example, the Venn diagram represents the intersection of barFLEX and YETI has $62 + 111 = 173$ but the panel b says "all shared" are 62 and barFLEX and YETI are 111. Did you exclude the "all shared" from barFLEX and YETI in the panel b? What is the point of showing both of the panels? Furthermore, rather than redefining the quantitative growth values into the discrete labels of "impaired growth" or not, the authors should show scatter plots of growth values from different assays with statistical tests. Why was this avoided?

While the percentages in panel S8A and S8B (now called Figure EV3) are similar and therefore may appear redundant, they actually contain two different data types.

Panel S8A is comparing toxicity screens with barFLEX and GST to the YETI collection at 100 nM β -estradiol (the highest concentration tested on all strains). Toxicity was also measured at multiple β -estradiol concentrations (1, 5, 10, 100 nM) in YETI and allows us to infer which genes are toxic at low levels versus higher β -estradiol concentrations.

In panel S8B we use all measurements of toxicity to report the percentage of YETI strains toxic only at a high estradiol concentration (100 nM) versus lower concentrations (1, 5, 10 nM). The purpose of 8B is to understand how well each collection (YETI, GST, barFLEX) categorizes toxicity given the strength of toxicity. We therefore intentionally excluded "all shared" from the barFLEX and YETI comparisons so that the percentages better represent the degree of toxic genes determined in the barFLEX collection. The colors are coordinated between panels A and B.

We've updated the figure to clarify the issues the reviewer raises.

Finally, the reviewer suggests making scatterplots to compare different datasets. This is a good suggestion. However, we needed to use discrete values of "impaired growth" so that we could compare the YETI collection to multiple published toxicity datasets. The GST experiment does not provide the raw data but instead a list of toxic genes and the severity of the growth defect as categorical values. The barFLEX experiment does include raw data but direct comparisons would be difficult given the differences in experimental design. Therefore, we simply extracted the genes that were reported as toxic in the barFLEX paper.

2-9. Fig 5 is just a heatmap conversion of the raw data and doesn't provide much information. Also, many discussions here are based on data presented by large supplementary tables and hard to understand. Figure displays would greatly help the understanding of what the authors are claiming. For example: "The resulting data were normalized to data collected at time 0, then hierarchically clustered to look for general trends (Table S7A)" This doesn't represent any hierarchical clusters. "Indeed, we saw strong GO enrichment for biosynthetic processes, including amino acid and organic acid biosynthesis, and the average Aux score for genes in this cluster was relatively high (Table S7B)" This could be a small panel in Fig 5 with statistical tests.

We respectfully disagree with this comment from the reviewer about the visualization -- the heatmap shows the dataset in its totality and provides an immediate and holistic view of the data to the reader. The GO enrichments are quantified for clusters that we identified by hierarchical clustering and, beyond the synthesis of our findings in the main text, details/statistics are contained tidily in Dataset EV7. We have not included the dendrogram in the figure for simplicity (the clusters we chose are easy to see in the heatmap itself). In addition to providing the lists of genes in the labeled clusters, we've provided the clustergram/tree information in .CDT/.GTR format in the revision for the interested reader/analyst (Dataset EV11). A visualization can be seen below.

2-10. Fig S9: It looks like only a small fraction of the extremely enriched and depleted strains in the Bar-seq assay agreed with the on-plate data, and this assay could produce many false positives. If the Bar-seq assays of different estradiol concentrations and those of the on plate assays are compared, respectively, how do they look like?

Unlike plate-based assays, BAR-seq features competitive growth. Our intuition is that differences in growth rates between strains result in fold-change increases in barcode frequency that are amplified as compared to differences in plate-based growth measurements. This intuition is reflected in the larger dynamic range of fold-changes of barcodes versus AUGC values. We only performed BAR-seq at one inducer concentration. The purpose of the BAR-seq assay was to validate the barcodes and provide independent confirmation of our plate-based assays. We succeeded at both. In **Appendix Figure S8**, there are four quadrants (delineated by a grid). The majority of data is in the lower left quadrant, demonstrating broad agreement between the two assays. Our expectation is that the yeast community will generally use the BAR-seq protocol for functional genomics screens.

2-11. Page 12: "To explore this approach with the YETI collection, we chose ROF1, which encodes a putative transcription factor containing a WOPR DNA-binding domain and whose induction with \$\beta\$ -estradiol inhibits growth, both on plates and in BAR-seq experiments [73,74] (Figure 5, Figure 6, Table S6A, Table S7A)." Where is ROF1 in Fig 5? Why Fig 6 is cited here?

In Figure 5, *ROF1* (*YHR177W*) is contained in the “+e 1b” cluster. We’ve now labeled the figure to show where *ROF1* is. Using Java TreeView we show the *ROF1* barcode data below. The blue color indicates that the Z₃pr-ROF1 strain is becoming less abundant in the timecourses in which the pools of yeast cells are treated with 1 μ M β -estradiol.

We agree with the reviewer -- **Figure 6** should not have been cited here. Following the reviewer’s comment, we’ve updated the text to read, “To explore this approach with the YETI collection, we chose *ROF1*, which encodes a putative transcription factor containing a WOPR DNA-binding domain [74,75]. *ROF1* induction with β -estradiol inhibits growth (**Figure 5**) both on plates (**Dataset EV6A**) and in BAR-seq (**Dataset EV7A**) experiments.”

2-12. Page 13: "ROF1 also showed SDL interactions with numerous genes involved in chromatin remodeling and general transcription, including multiple members of the Swr1 complex, Ino80 complex, Rpd3L complex, NuA4 complex, and COMPASS, which presumably reflects its role as a transcriptional repressor and a regulator of other TFs." How do their transcriptomes look like? Do they agree with what the authors observed in Fig 6?

The reviewer is suggesting that downregulation of the same genes in chromatin remodeler mutants and from *ROF1* overexpression could lead to a severe growth defect, explaining the SDL interactions. Although we saw downregulation of some genes in the *ROF1*-repressed regulon, there was no consistent pattern among all the chromatin remodelers. We add a sentence: *Comparison of the ROF1 overexpression microarray profile to microarray profiles of strains deleted for chromatin regulators did not reveal any obvious pattern of co-regulation that might provide a mechanism for the SDL interactions (Lenstra PMID: 21596317).*

We did not perform any transcriptome experiments with strains containing mutations in genes with which Rof1 has SDL interactions. Based on this question from the reviewer, we tested if the genes that Rof1 has the strongest SDL interactions with are also repressed by Rof1 in the gene expression time course. None of the genes with the 25 strongest SDL scores were repressed by Rof1.

2-13. Page 13: "These data suggest that many essential genes under the control of Z3EV may continue to be expressed in the absence of \$\beta\$ -estradiol for numerous cell doublings." Are they also lowly expressed genes?

It is hard to make generalizations from 24 strains. However, we compared the expression levels of the 16 genes that were not reversible and of the 8 genes that showed at least some reversibility. There was no statistical difference in native expression levels between these two groups of genes. We've added the following text: *"There was no difference in native expression between the 16 genes that lacked reversibility and the 8 genes that showed either full or partial reversibility (p -value = 0.35, two-sided Student's t -test)."*

2-14. Re: "reversibility," is there any epigenetic memory effect observed? Like, do some epigenetic memories get triggered by overexpression of a target gene, which do not allow the cells to revert back even after removing estradiol or affect the secondary induction? What I heard about this collection a while ago, I thought one strong application would be a genome-wide study of such epigenetic memory effects.

This is an interesting point and could be the subject of future research. We know that ZEV induction (like in the case *ROF1* shown here) results in transcriptional changes, as well as changes to chromatin structure (Hendrickson et al., *Methods in Molecular Biology*, 2018). Changes in chromatin structure that persist could, in our minds, result in an epigenetic memory of past events. We've added the following text: *"We previously found that changes in DNA accessibility using ATAC-seq are correlated with future changes in RNA expression [31]. It is possible that increased DNA accessibility at Z₃pr following β -estradiol addition may not be easily reversed upon β -estradiol removal, and the level of phenotypic reversibility depends on multiple gene-dependent variables."*

Minor comments:

2-15. Page 4: "To select for strains that carried a wild-type HAP1 gene, the gene encoding the Z3EV transcription factor was integrated next to the repaired HAP1 together with a natMX selectable marker in Y14789." What do you mean by "repaired"? Doesn't Y14789 encode a functional HAP1?

We have removed the word "repaired", and now use the word "functional" to describe the *HAP1* allele without the transposon insertion.

*"We chose RCY1972 as it is deleted for the HIS3 locus, making it compatible with SGA methodology, but otherwise prototrophic, enabling studies of yeast cell growth and other phenotypes in a variety of conditions [51]. The strain also carries a **functional** HAP1 gene, which encodes a transcription factor that localizes to both the mitochondria and nucleus and is required for regulation of genes involved in respiration and the response to oxygen levels [52]. Strains derived from S288C **typically** carry a Ty1 element insertion in the 3' region of the HAP1 coding sequence, creating a HAP1 allele that acts as a null for cytochrome c expression and leads to mitochondrial genome instability [52]. Previous work has shown that removal of the Ty element, which repairs the HAP1 gene, increases sporulation efficiency dramatically [53]. To select for strains that carried a **functional** HAP1 gene, the gene encoding the Z₃EV transcription factor was integrated next to HAP1 together with a natMX selectable marker in Y14789."*

2-16. Y14789 has a different genetic background from the BY strain, the background of the deletion collections. Is this problematic for genetic interaction assays using the SGA methodology?

We state in the Results, p4, Our diploid parental strain, Y14789, was based on the RCY1972 strain, **a derivative of S288C**. We chose RCY1972 as it is deleted for the *HIS3* locus, making it compatible with SGA methodology.

Y14789 and standard BY strains are closely related S288C derivatives (they are in the same background, except we've added a functional *HAP1* allele as described in Hickman and Winston [Molecular and Cellular Biology, 2007]). Y14789 is prototrophic for *LEU2*, *MET15* and *LYS2*. We completed a Synthetic Dosage Lethality (SDL) screen with *ROF1* in part to show that the collection is indeed compatible with SGA methodology.

2-17. I thought it would be nicer if authors encode a transcribing DNA barcode (RNA barcode) under the Z3 promoter such that the gene induction levels can be massively quantified by deep sequencing of RNA barcodes and their genomic DNA.

We agree that this is a good idea, but we didn't incorporate RNA barcodes into the collection. In the Discussion, we note: *"To make the YETI collection compatible with perturb-Seq-like approaches for monitoring regulatory networks at the single-cell level, barcodes encoded in RNA could be introduced [89]."*

2-18. Having a discussion on their system in contrast to CRISPRa would be nice. Some people would quickly think constructing a gRNA library for CRISPRa is way easier than creating these strain libraries.

We've added the text in red below to the discussion to explain that the YETI enables a particular experimental design that is not obtainable with CRISPRa. *"We constructed and characterized YETI, a genome-scale collection of Z₃EV inducible alleles of yeast genes. Previously, we established that combining inducible expression with transcriptome-wide time series measurements is an effective strategy for elucidating gene regulatory networks. The ability to switch a gene on and measure dynamically how every other gene responds has allowed us to identify causal regulatory interactions, both direct and indirect, and observe instances of feedback control that were previously inaccessible (Mclsaac et al, 2012; Hackett et al, 2020). **Furthermore, the experimental design facilitated by this collection, in which a gene is switched from off to on, cannot be achieved with methods like CRISPRa** (Gilbert et al, 2014). By creating the YETI collection, we are expanding the uses of this system from studies of individual genes to nearly all genes in the yeast genome."*

2-19. Page 13: "However, 16 of the 24 (67%) Z3EV strains grew in the absence of \$\beta\$ -estradiol, even though their initial growth had been \$\beta\$ -estradiol-dependent." Fig S10 needs to be cited here.

We thank the reviewer for pointing out this oversight. The text now reads, "*However, 16 of the 24 (67%) Z₃EV strains grew in the absence of β -estradiol, even though their initial growth had been β -estradiol-dependent (Appendix Figure S9).*"

Reviewer #3:

In this paper, the authors generate a genome-wide collection of yeast strains. Each of these strains contains an inducible promoter swapped substituted for a different gene's native promoter. These inducible promoters are ZEV variants that are inducible with estradiol. This collection is a powerful and versatile resource because yeast is not naturally responsive to estradiol, meaning genes can be induced in different environments and without triggering a general response to the estradiol. This whole system is much more precise and titratable than alternatives, such as galactose induction.

This paper is a nice achievement, as generating and checking all of these strains was undoubtedly a substantial amount of work. This collection seemed to have been produced in a careful and rigorous manner. The experiments that are described show how interesting biological insights can be obtained using the YETI collection. I should also say, this paper was written with simple and easy to read language. I felt anyone could read and mostly understand this manuscript, which will help ensure this paper has a broad audience.

We appreciate the positive feedback, and are pleased that the reviewer thought the paper will be accessible to a broad audience.

I do not have any substantive comments on this paper. The following suggestions are more nitpicky and are simply places where I noticed a bit more information or clarity might be helpful.

3-1. p4, RCY1792. Is this an S288c derivative? The name was unfamiliar and it would be good to connect this to the commonly used strains if possible.

We've clarified the text, and added, "*Our diploid parental strain, Y14789, was based on the RCY1972 strain, an S288C derivative.*" This is a standard S288C-derived strain with a functional *HAP1* allele (i.e., no transposon insertion).

3-2. p5, 'TPS2 or LEU2' should probably be flipped to 'LEU2 or TPS2' given what comes later in the sentence.

We thank the reviewer for pointing this out. We've adjusted the text to read, "*We engineered strains in which LEU2 or TPS2 were placed downstream of Z₃pr and in the absence of β -estradiol,*

the resultant strains displayed known phenotypes associated with leu2Δ (Appendix Figure S1B) and tps2Δ (Appendix Figure S1C), leucine auxotrophy (Toh-e et al, 1980) and heat sensitivity (Gibney et al, 2015), respectively, but grew equivalently to WT cells in the presence of β-estradiol.

3-3. p5. The integration of the Z3 promoters is described only superficially in the main text. Details are in the methods. However, a bit more information is probably necessary in the main text.

We agree with the reviewer that more explanation in the main text would be helpful for readers. We've modified the main text accordingly:

“The URA3 gene marking the Z₃ promoter is expressed divergently from the Z₃pr-controlled target gene. Importantly, the URA3 marker gene in each strain is linked to a unique DNA molecular barcode such that the resulting genome-wide β-estradiol-inducible strain collection is compatible with pooled screening approaches (Figure 1A). “Promoter insertions were placed between the first ATG of each ORF and did not remove any native DNA. Rather than removing the native promoter sequence from the genome, which we believe is likely to disrupt the expression of neighboring genes, native promoters were simply displaced by ~2kb. Additionally, yeast does not have “transcriptional activation at a distance.” From the work of Dobi et al., once an activation sequence is >700 bp from a target gene, it is no longer regulatory [57]. Thus, we expect that displacement of the native promoter by ~2kb should be sufficient for removing its regulatory potential.”

3-4. p8, The E. coli discussion was interesting, but is there no comparable data in yeast? For example, the Steinmetz lab has done genome-wide CRISPRi in yeast.

We obtained similar results for four different yeast expression technologies: Tet-off, Temperature sensitive degrons, DAmP alleles, and CRISPRi. Data are shown below from our original Figure S4 (which is now integrated into **Figure 3**). Given the comments from the reviewers, we've moved some of these data to the main text.

3-5. p9, 'strong selection pressure resulting in aneuploidy'. The aneuploidies happen and are then selected. It is not that selection generates aneuploidies, unless there is something I am missing.

We agree that our phrasing was confusing and we have removed the offending text.

3-6. p14, The main text might benefit from a brief description of why reversibility might occur.

We have added additional text speculating on why growth reversibility may occur for some essential genes and not others with the Z₃EV system (see response to reviewer comment 2-14).

3-7. f5, The panel at the top is a little confusing in my opinion. Is there a more intuitive way to draw the sampling scheme for Barcode sequencing?

We have simplified the sampling scheme for Figure 5. We show the modified figure below:

a BAR-seq experimental design

4 experimental conditions

		β -estradiol	
		0 nM	100 nM
SC	SC+e	Synthetic Complete	
YNB	YNB+e	Yeast Nitrogen Base	

Manuscript Number: MSB-2021-10207R, A genome-scale yeast library with inducible expression of individual genes

Thank you for sending us your revised manuscript. We have now heard back from reviewer #1 who was asked to evaluate your revised study. As you will see below, the reviewer is satisfied with the modifications made and supports publication. They only raise two rather minor points, which we would ask you to address in a minor revision.

We would also ask you to address some remaining editorial issues listed below.

Reviewer #1:

The authors have addressed my comments satisfactorily. I have two follow-up notes related to specific things that were changed in revision, but both are very minor:

1. The added statement "Furthermore, the experimental design facilitated by this collection, in which a gene is switched from off to on, cannot be achieved with methods like CRISPRa" does not seem to be supported by Gilbert 2014 (and in fact a doxycycline-inducible CRISPRa system was published in PMID 31959800). If there is some specific basis for this statement it should be made clear.

2. The added EV2 has colony size (and the associated AUGC metric) in units of pixel². My understanding is that pixels are a unit of area, and I have no intuition as to what a pixel-squared is. I would ask the authors to clearly explain what this unit is in the methods section and how the reader should interpret it.

Reviewer #1:

The authors have addressed my comments satisfactorily. I have two follow-up notes related to specific things that were changed in revision, but both are very minor:

1. The added statement "Furthermore, the experimental design facilitated by this collection, in which a gene is switched from off to on, cannot be achieved with methods like CRISPRa" does not seem to be supported by Gilbert 2014 (and in fact a doxycycline-inducible CRISPRa system was published in PMID 31959800). If there is some specific basis for this statement it should be made clear.

We agree with the reviewer, and didn't make our point clearly. We've replaced the offending text with "*All engineered components are directly integrated in the genome (i.e., no plasmids), a feature that is especially useful for achieving homogeneous expression of individual target genes.*"

2. The added EV2 has colony size (and the associated AUGC metric) in units of pixel². My understanding is that pixels are a unit of area, and I have no intuition as to what a pixel-squared is. I would ask the authors to clearly explain what this unit is in the methods section and how the reader should interpret it.

We thank the reviewer for catching these typos. The y-axis now reads "Colony size (# of pixels)" for each plot.

RE: MSB-2021-10207RR, A genome-scale yeast library with inducible expression of individual genes

Thank you again for sending us your revised manuscript. We are now satisfied with the modifications made and I am pleased to inform you that your paper has been accepted for publication.

Corresponding Author Name: R. Scott McIsaac

Manuscript Number: MSB-2021-10207